# How to Identify Potentials and Barriers of Raw Materials Recovery from Tailings? Part II: A Practical UNFC-Compliant Approach to Assess Project Sustainability with On-Site Exploration Data

Rudolf Suppes [1,2,*] and Soraya Heuss-Aßbichler [3]

1   Institute of Mineral Resources Engineering (MRE), RWTH Aachen University, Wüllnerstr. 2, 52064 Aachen, Germany
2   CBM GmbH—Gesellschaft für Consulting, Business und Management mbH, Horngasse 3, 52064 Aachen, Germany
3   Department of Earth and Environmental Sciences, Ludwig-Maximilians-Universität München, Theresienstr. 41, 80333 Munich, Germany; soraya@min.uni-muenchen.de
*   Correspondence: rudolf.suppes@rwth-aachen.de or suppes@cbm-ac.de

**Abstract:** A sustainable raw materials (RMs) recovery from waste requires a comprehensive generation and communication of knowledge on project potentials and barriers. However, a standardised procedure to capture sustainability aspects in early project development phases is currently missing. Thus, studies on different RM sources are not directly comparable. In this article, an approach is presented which guides its user through a practical interpretation of on-site exploration data on tailings compliant with the United Nations Framework Classification for Resources (UNFC). The development status of the overall project and the recovery of individual RMs are differentiated. To make the assessment results quickly comparable across different studies, they are summarised in a heat-map-like categorisation matrix. In Part I of this study, it is demonstrated with the case study tailings storage facility Bollrich (Germany) how a tailings mining project can be assessed by means of remote screening. In Part II, it is shown how to develop a project from first on-site exploration to a decision whether to intensify costly on-site exploration. It is concluded that with a UNFC-compliant assessment and classification approach, local sustainability aspects can be identified, and a commonly acceptable solution for different stakeholder perspectives can be derived.

**Keywords:** anthropogenic raw materials; sustainability assessment; tailings recycling

## 1. Introduction

A growing world population, the growth of emerging economies, and the global transition to a decarbonised energy supply lead to an increasing demand for mineral raw materials (RMs) [1–4]. For more than a century, the annual average increase in global mineral RM demand is reported to be 3% [1], and a 2- to 3-fold increased global demand for Al, Cu, Fe, Mn, Ni, Pb and Zn is expected between 2010 and 2050 [5,6]. Due to net stock additions and low recycling rates, the primary mining industry is expected to remain an important supplier of RMs in the foreseeable future [6,7].

In mining, valuable RMs are extracted from ores by separating wanted from unwanted minerals. A common method to do so is froth flotation, which requires the ores to be finely ground to a particle size of typically 10–200 μm [8]. The unwanted minerals are rejected as tailings, and they are usually stored in tailings storage facilities (TSFs). The global annual tailings production is estimated to lie in the range of 5–14 Gt [9], and it is estimated that in China alone some 12,000 TSFs exist [10]. Globally, ore grades are decreasing and ore complexities are increasing [11] so that the amount of produced tailings and energy spent per unit of produced commodity are increasing.

Despite continuous improvements in the construction and management of TSFs, they can be regarded as legacies with long-lasting environmental impacts, such as the occupation of large surface areas, and high external costs [12–16]. Risks associated with TSFs comprise the contamination of soil and water with acidic leachates or heavy metals, especially in the case of sulphidic tailings [13,17–19]. Other risks include dam stability issues which, on average, cause 2 to 3 annual TSF failures, leading to a contamination of large areas and threatening human lives [20,21]. The environmental impact of TSFs has increased public pressure on the primary mining industry to act more environmentally friendly [6,22,23].

At the same time, tailings contain usable RMs due to former processing inefficiencies or an emerging demand for RMs which were not exploitable in the past [24]. The active promotion of sustainability in RM sourcing in the past decade by institutions such as the European Commission (EC) has initiated a paradigm shift so that formerly regarded waste is now becoming interesting for valorisation [25–27]. Scientists have investigated the recovery of metalliferous or industrial minerals from tailings [28–30], or an alternative valorisation, e.g., in construction materials [31–33] or glass making [34–36].

A comprehensive exploration is required to identify if tailings can be valorised. However, conventional case studies under consideration of the Committee for Mineral Reserves International Reporting Standards (CRIRSCO) classification principles from the primary mining industry usually target single RMs and neglect other contained RMs (cf., References [37–39]). Hence, the knowledge on their RM potential is incomplete. Usually, economic aspects are mainly considered in the primary mining industry [8,40], while environmental and social aspects of RMs recovery are mostly neglected or ignored; only recently have sustainability aspects been given greater attention [41].

The United Nations Sustainable Development Goals aim at a worldwide sustainable extraction of natural RMs [42]. Therefore, the prospects of mineral RMs recovery requires environmental and social aspects to be regarded as equal to economic ones. As a result, these aspects must be assessed concurrently with geological, technological, and legal aspects to obtain comprehensive exploration results [43]. This is possible when applying the United Nations Framework Classification for Resources (UNFC) principles, which are based on the 3 categories: *degree of confidence in the estimates* (G category), *technical feasibility* (F category), and *environmental-socio-economic viability* (E category) [44]. In this way, decision-makers in RM management can get an overview of the potentials and barriers of mineral RMs recovery from tailings and its competitiveness across different RM sources.

In mineral RM exploration in the primary mining industry, a mineral deposit is first identified with remote techniques [8,45]. It is then investigated on site with intensified techniques to obtain data for a first techno-economic assessment, termed a scoping study [8,45]. Despite the many recent case studies on anthropogenic RMs developed in analogy to natural RMs [46], a standardised procedure is missing. Existing case studies provide a snapshot of a specific stage of project development in the RMs recovery chain [47], e.g., the remote exploration [48]. Hence, there is a research gap in the development of case studies which outline the progression of RMs recovery project development [47].

This study addresses the lack of a standardised procedure to explore tailings as anthropogenic RMs. It is the first to demonstrate how a UNFC-compliant tailings mining project assessment and classification can evolve from a first remote TSF screening (Part I [43]) to a consecutive interpretation of on-site exploration data (Part II). In this article, a systematic and practical UNFC-compliant approach is developed for a very preliminary assessment and classification of tailings mining projects based on on-site exploration data. It is tested to what extent an overview of project potentials and barriers can be obtained. The research questions are: (1) is it possible to reconcile different stakeholder interests with a UNFC-compliant approach or must different perspectives be considered on their own merits? (2) which aspects should be considered in very preliminary UNFC-compliant assessments? (3) can a UNFC-compliant approach be used to identify site-specific project potentials and barriers?

The approach focuses on metalliferous tailings from industrial processes. A project's development status is differentiated in terms of geological, technological, economic, environmental, social, and legal aspects. Beside the rating of the overall project, each contained RM is rated individually as a separate subproject. The rating is performed in a categorisation matrix in a heat map-like style. In this way, driving factors as well as barriers can be identified quickly. The approach is tested with the case study TSF Bollrich (Germany) from a public decision-maker's perspective, considering the interests of local environmental non-governmental organisations (NGOs), private investors, and the city administration of Goslar. The TSF was chosen since it is a potential source of economically highly relevant RMs, it is situated in a complex environment with several stakeholders, and there is a potential to relieve the burden on the environment and society [43].

The article is structured as follows: (i) outline of the frame conditions for the further development of the case study Bollrich, (ii) proposal of a UNFC-compliant anthropogenic RMs assessment and classification approach, (iii) development of a categorisation matrix for a UNFC-compliant rating of the overall project and subprojects for individual RMs, (iv) case study application, and (v) discussion of the developed approach.

## 2. Terms and Methods

### 2.1. Key Words and Definitions

*TSF:* physical structure for tailings storage. *Deposit*: potential RM source. *Target minerals:* minerals wanted for valorisation. *Other minerals*: unwanted minerals. *Recovery:* physical extraction process. *Material recovery:* extraction of minerals to be used in construction materials. *Tailings mining:* process from exploration, recovery, and processing to rehabilitation. A *very preliminary study* is regarded as an analogue to a scoping study from the primary mining industry [45] (p. 31), and it is defined as follows: *it is the first quantification of a tailings mining project's potentials and barriers with respect to geological, technological, economic, environmental, social, and legal aspects. The degree of uncertainty in the estimates is high. The study is based on directly generated project data, for instance from on-site exploration or information from other sources such as from the literature and model assumptions based on similar projects. Technological considerations are based on conceptual foundations.*

### 2.2. Considerations for the Development of the Case Study TSF Bollrich

This case study is based on the screening results from Reference [43], where the following potentials are identified: an economic interest in the TSF is justified due to its size and the presumably contained critical raw materials (CRMs) $BaSO_4$ and In, as well as the highly economically relevant RMs Ag, Au, Cu, Pb, and Zn. The development costs are expected to be low since buildings, transportation, and utilities infrastructure are present in the near vicinity. As Germany has a high rating on the ease of doing business ranking, favourable regulatory conditions for an investment can be assumed. The TSF's environment is vulnerable to a potential TSF failure: the nearest human settlement is located ~400 m downstream of the TSF, and the high score on the Human Footprint Index indicates that land-use-related social tension with competing interests can be expected in the area. Therefore, a removal of the TSF would reduce the potentially severe risks of a TSF failure.

The following barriers are identified [43]: the TSF is located in a challenging environment with a potential for social conflicts due to agricultural, forest, industrial and commercial, nature and water protection, recreation, and residential areas in the near vicinity. A diverse and socially active stakeholder group of a minimum of 18 parties could be identified, which may potentially form a strong base for a project rejection. Amongst others, these include environmental NGOs, the Development Association Cultural Heritage Ore Mine Rammelsberg, and the Air Sports Community Goslar. The geological knowledge on the deposit is limited due to unknown RM quantities and qualities. Furthermore, potentially contained RMs are presumed based on literature on mined ores and their processing. Knowledge on the TSF's geomechanical stability is missing. Valuable ecosystems with

protected species have formed as a result of ecological succession. To overcome these barriers, on-site exploration and evaluating techno-economic feasibility is required; local stakeholders' environmental, social, and economic interests must be considered; and advantages and disadvantages of RMs recovery need to be weighed against each other.

*2.3. UNFC-Compliant Anthropogenic Raw Materials Assessment and Classification Approach*

The assessment and classification approach from Heuss-Aßbichler et al. [47] (p. 17) was adopted and modified by adding sub-steps and assigning assessment methods. The modified approach consists of 3 phases (cf., Figure 1), which can be reiterated when additional information is required or when new information on preceding steps is generated:

1.　Definition of project and generation of information.
2.　Assessment of project's development status.
3.　UNFC-compliant categorisation of criteria and project classification.

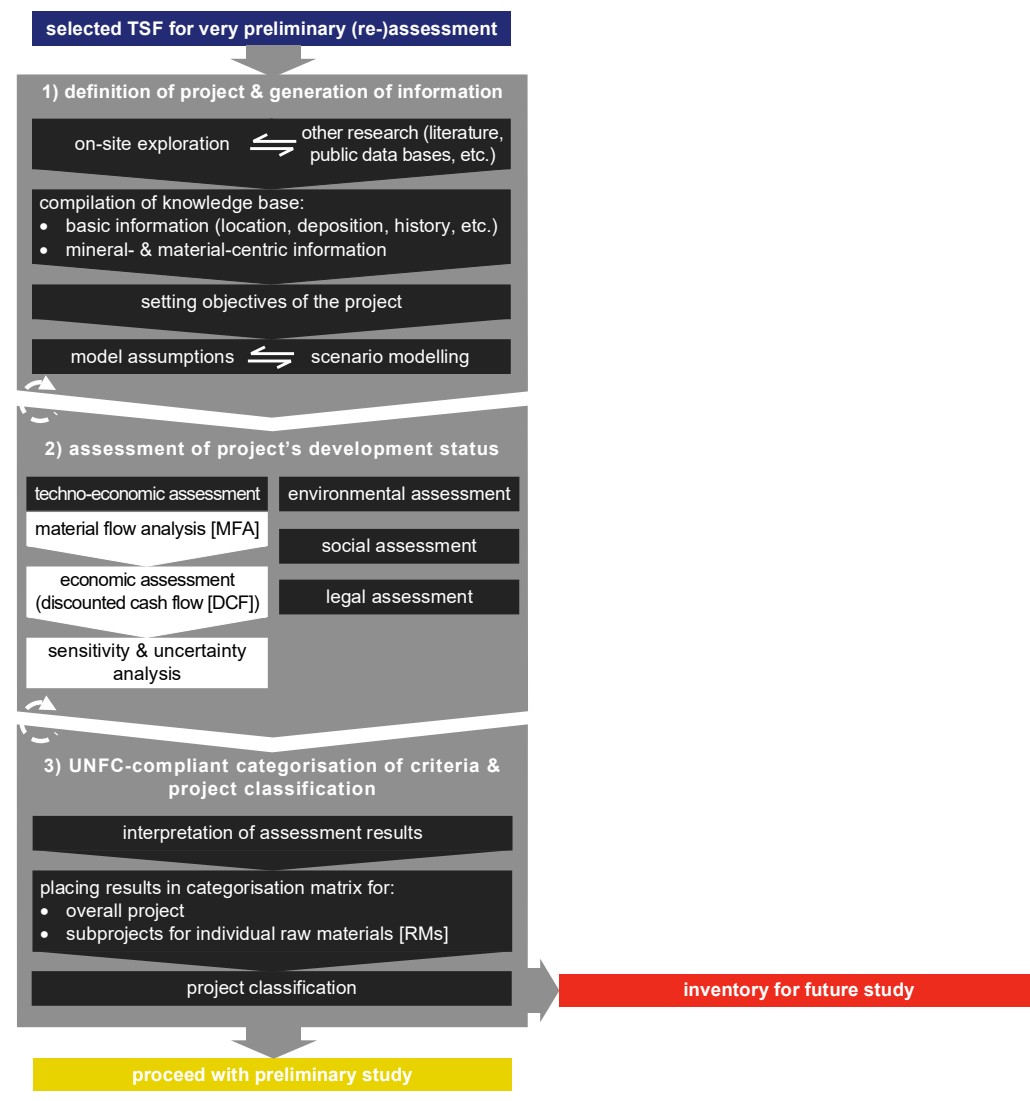

**Figure 1.** Practical UNFC-compliant approach for a systematic assessment and classification of mineral RMs recovery from tailings at very preliminary level. The leftwards arrow over rightwards arrow indicates mutual influence, and the dotted circles indicate possible reiteration steps.

*2.4. Case Study Assessment Methods*

2.4.1. Environmental Assessment

TSF-related risks can have a great influence on the classification result of a tailings mining project [49]. Based on data from scientific literature, publicly accessible sources, and observations on Google Earth [50], a status quo risk assessment is performed. The TSF's stability and its impacts on the surrounding environment is assessed, including the following subjects of protection (adopted from Reference [51]): air, flora and fauna, ground, groundwater, human health, landscape, and surface water.

2.4.2. Social Assessment

Investors are recognising that ignoring social aspects in project development can create barriers to RMs recovery [6]. Amongst others, it is therefore important to consider the attitudes of local stakeholders such as communities towards a possible RMs recovery. From the stakeholders identified in Reference [43], this study focused on administrative bodies, industry, and local environmental NGOs as proxies for concerned citizens. Due to a lack of data, only basic tendencies on stakeholder attitudes are assessed. The assessment is based on an internet search and the study of Bleicher et al. [52] who interviewed stakeholders on a potential RMs recovery from mine waste in the Harz region including the TSF Bollrich. They focused on stakeholders from non-specified local and regional environmental NGOs, industry, administrative bodies, and scientific institutions, and they considered secondary sources such as public media.

2.4.3. Material Characterisation and Material Flow Analysis

The drill core sampling campaigns on the TSF Bollrich for tailings characterisation are described in References [53,54]. 3 scenarios are developed: no RMs recovery (NRR0), conventional RMs recovery (CRR1), and enhanced RMs recovery (ERR2). The amount and composition of generated commodities and residues are evaluated with a material flow analysis (MFA) according to Reference [55] under consideration of available recovery technologies:

1. Scenario definition and selection of relevant processes and mass flows.
2. Mass flow quantification with published and estimated data, and model assumptions for unavailable data.
3. Mass flow visualisation with Sankey diagrams.

2.4.4. Economic Assessment

The economic viability is assessed with a discounted cash flow (DCF) analysis to determine the net present value (NPV) before taxes, considering internal costs and revenues. The NPV is estimated with the open-source software R (www.r-project.org, accessed on 16 January 2021) after

$$NPV = -I_0 + \sum_{i=1}^{t} \left( I_i / (1+r)^r \right), \tag{1}$$

where $I_0$ is the initial investment [€] in year 0, $I_i$ is the net cash flow [€] in the *i*-th year, $r$ is the discount rate [-], and $t$ is the project's duration [a]. Given estimated figures for target mineral masses, prices and recovery rates are rounded down; they are rounded up for costs to estimate conservatively as per CRIRSCO [45].

2.4.5. Sensitivity and Uncertainty Analysis

To increase the reliability of the assessment, sensitivity and uncertainty analyses is performed [56]. The sensitivity analysis is performed by varying input factors to determine how the outputs depend on them. The uncertainties are assessed with dynamic price forecasts by applying autoregressive functions to historical price data of metals, minerals, diesel, and electric energy (cf., Supplementary Materials, Figures S1–S9).

### 2.4.6. Legal Assessment

The legal aspects right of mining, environmental protection, and water protection are considered. Due to a lack of data, the state of development of legal aspects are assessed by making basic considerations based on data from Reference [53].

### 2.5. Development of a Categorisation Matrix for a UNFC-Compliant Project Rating

In the categorisation matrix, the overall project and subprojects for individual RMs are differentiated. The UNFC's G, F, and E categories are addressed. The E category is subdivided into economic (a), environmental (b), social (c), and legal (d) aspects, the latter being defined as a distinct subcategory in this article. For the project categorisation and classification, an exemplary 35 factors for the rating of the overall project and 9 factors for the rating of the subprojects for individual RMs are assessed. They are adapted and modified after a literature search on established assessment factors from the primary mining industry, literature on sustainability in mining, case studies, and our own reasoning. Table 1 provides an overview of the chosen factors, their allocation to groups, and the rationale for choosing them based on their influence on a project. A proposal is made for a UNFC-compliant rating with *descriptive indicators* to describe a state and *performance indicators* to quantitatively compare the status quo with target values. For better legibility, the categorisation matrix is divided into separate tables (cf., Appendix A, Tables A1–A10). With the above nomenclature, an exemplary rating in the social subcategory might look like E3.1c or E1c. Factors with high uncertainty remain in the 3rd UNFC subcategorisation (3.1, 3.2, 3.3), while more developed factors can be rated as high as in the 1st UNFC category (1, 2, 3). For a quick overview of project potentials and barriers, an individual colour is assigned to each rating. In the discussion in Section 4.1, the rating results are presented in a heat-map-like style for a quick overview.

**Table 1.** Categorisation matrix: assessed factors and rationale behind their application based on their influence on a project.

| Category & Factor | Influence on | UNFC Axis [1] |
|---|---|---|
| **overall project rating** | | |
| *geological conditions (relevant for project development)* | | |
| (1) quantity, (2) quality, (3) homogeneity | potential profitability, mine planning, overall uncertainty | G |
| *TSF condition & risks (relevant for project development)* | | |
| (4) ordnance | exploration costs, overall project safety | F |
| *mine planning considerations (relevant for project execution)* | | |
| (5) mine/operational design, (6) metallurgical testwork, (7) water consumption | reliability of the financial analysis, efficiency of the operation, environmental footprint | F |
| *infrastructure (relevant for project development)* | | |
| (8) real estate, (9) mining & processing, (10) utilities, (11) transportation & access | project viability, ramp-up time | F |
| *post-mining state (relevant for future impacts)* | | |
| (12) residue storage safety, (13) rehabilitation | necessary aftercare measures, public acceptance | F |
| *microeconomic aspects (relevant for project development)* | | |
| (14) economic viability, (15) economic uncertainty | potential returns, investor interest | E a |
| *financial aspects (relevant for project development)* | | |
| (16) investment conditions, (17) financial support | potential returns, investor interest, security of investment | E a |
| *environmental impacts during project execution* | | |
| (18) air emission, (19) liquid effluent emission, (20) noise emission | mine planning, local population, local ecosystems | E b |
| *environmental impacts after project execution* | | |
| (21) biodiversity | quality of ecosystem after the project | E b |
| (22) land use | land which can be repurposed | |
| (23) material reactivity | aftercare measures, local ecosystems | |
| *social impacts during project execution* | | |
| (24) local community, (25) health & safety, (26) human rights & business ethics | social acceptance, peace & wellbeing, (unforeseeable) costs for compensation | E c |

**Table 1.** *Cont.*

| Category & Factor | Influence on | UNFC Axis [1] |
|---|---|---|
| *social impacts due to project execution* <br> (27) wealth distribution, (28) investment in local human capital <br> (29) degree of RM recovery, (30) RM valorisation | social peace & wellbeing, employment of local population, valuable legacy for workers & society after mine closure amount of new residues, ecological risks, effort for & efficiency of future RMs recovery | E c |
| *social impacts after project execution* <br> (31) aftercare, (32) landscape | social risks, social wellbeing, external costs | E c |
| *legal situation (relevant for project development)* <br> (33) right of mining, (34) environmental protection,(35) water protection | project feasibility, social acceptance, effort for formal project planning | E d |
| subproject for individual RMs rating | | |
| *geological conditions (relevant for project development)* <br> (36) quantity, (37) quality, (38) homogeneity | potential profitability, mine planning, RM uncertainty | G |
| *mine planning considerations (relevant for project execution)* <br> (39) recoverability | efficiency of the operation, amount of new residues | F |
| *microeconomic aspects (relevant for project development)* <br> (40) demand, (41) RM criticality, (42) price development | project viability, investor interest, overall project risk | E a |
| *impacts after project execution* <br> (43) solid matter, (44) eluate | environmental risks of new deposition, aftercare measures | E b |

[1] a: economic aspects, b: environmental aspects, c: social aspects, d: legal aspects.

## 3. Results

### 3.1. Definition of the Project and Generation of Information

3.1.1. Knowledge Base on the Case Study Deposit

The tailings deposit Bollrich (cf., Figure 2) near Goslar was part of the Rammelsberg mining operation [57]. It contains $BaSO_4$, Co, Ga, and In, which are CRMs in the European Union (EU), and the elements Cu, Pb, and Zn, which are economically highly important in the EU [58]. The deposit is nationally relevant as it is one of the few possible CRM sources [59]. The first exploration with a focus on geological aspects took place in 1983 before its abandonment in 1988 after ca. 50 years of operation [54]. In the 2010s, the exploration's main focus was on mineral processing. Geological, technological, environmental, legal, [53] and social aspects [52] were also investigated. A comprehensive assessment of a potential tailings mining project has not been carried out.

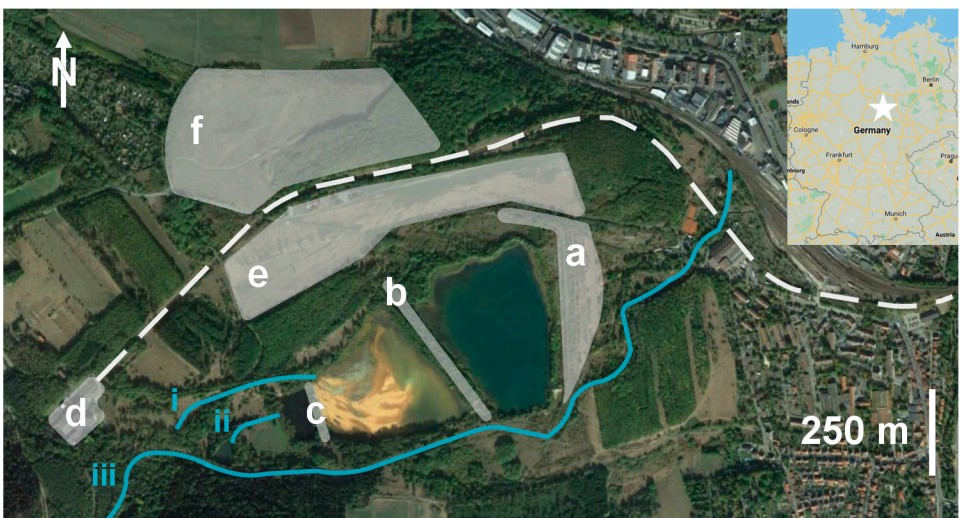

**Figure 2.** Schematic illustration of the TSF Bollrich's near environment: (a) marks the main dam, (b) the middle dam, (c) the water retention dam, (d) the disused processing plant, (e) a glider airfield, and (f) the disused landfill Paradiesgrund. The neutralisation sludge between the dams (b, c) is yellowish. The white dotted line marks the disused railway connection from Oker to the processing plant, (i) the stream of neutralised mine water, (ii) the connection between the pond Gelmketeich and the water retention pond, and (iii) the river Gelmke. Adapted after Google Earth [50].



In this study, the deposit in its current condition is assessed and classified from a sustainability viewpoint, considering the area around the TSF within a radius of 10 km. Information was derived from the existing scientific studies on the deposit in References [52–54,60] and from publicly available data sources. The knowledge base on the deposit is summarised in Table A11. The material flows and economics are evaluated quantitatively based on published data and model assumptions for unavailable data (cf., Table 2).

**Table 2.** Summary of model assumptions for the case study TSF Bollrich.

| Model Assumption |
| --- |
| (1) for in-situ rehabilitation, TSF abandonment is performed as for DK II class landfills [1] under the German Landfill Regulation (DepV) [61]. |
| (2) mass of dam material is neglected in mineral RMs recovery scenarios alongside its further treatment. |
| (3) freight costs for commodities & residues to downstream processes are neglected. |
| (4) all equipment can be used over the whole life of mine (LOM) without renewal except for the pipelines & pumps, which are exchanged in year 6 of the mining operation due to abrasive wear. |
| (5) processing plant Bollrich: assets can be used (for operation, administration, etc.), processing machinery can be reactivated, & the $BaSO_4$ concentrate can be conditioned on site; basic infrastructure is in place. |
| (6) experimental tailings recovery rates from lower pond applicable to tailings from upper pond, neglecting the influence of neutralisation sludge on processing. |
| (7) no losses & dilution of tailings occur during mining & transport. |
| (8) the processing plant produces 3 types of products: (i) a pure industrial mineral concentrate ($BaSO_4$), (ii) a mixed sulphide concentrate ($CuFeS_2$, $PbS$, $ZnS$) including all high-technology metals (Co, Ga, In), & (iii) mixed residues due to inefficiencies in mineral processing. |
| (9) smelters pay for the recoverable Co, Ga, & In content in the mixed sulphide concentrate based on a recovery with ammonia leaching as specified in Reference [60]. |
| (10) a discount rate of 15% is chosen to reflect a high risk investment [8]. |

[1] Above-ground landfill for contaminated but non-hazardous waste such as pre-treated domestic waste or commercial mineral waste. Geological base and surface sealing is required.

### 3.1.2. Setting Objectives of the Project

Based on current research, the TSF Bollrich offers the potential for action by a public decision-maker at national level seeking a sustainable solution at reasonable costs. Based on the stakeholder considerations (cf., Section 3.2.2), 3 relevant stakeholder perspectives are considered: NGOs with environmental concerns due to TSF-related risks, private investors seeking economic opportunities, and the city administration of Goslar seeking an opportunity to create high-value jobs and to establish a regional recycling industry.

The selected scenarios' objectives are: no RMs recovery (NRR0)—a physically and chemically stable, maintenance-free structure is created. Environmental and social risks are minimised by preventing the release of contaminants due to recovery and by avoiding the transport of hazardous material in a vulnerable region. The environment is rehabilitated, and the current landform is retained. RMs recovery (CRR1)—application of conventional technologies with off-site residue disposal. The original landform is restored, and the area is rehabilitated. RMs recovery (ERR2)—the same processes as in CRR1 but the produced residues are sold to a local recycling company.

### 3.1.3. Scenario Modelling

In the rehabilitation scenario (NRR0), a leachate collection system is installed, the TSF is stabilised by in-situ concrete injection, its surface is sealed, and leachates are captured and treated on site in a 5-year closure phase. In a 30-year aftercare phase, emissions and the

TSF's stability are monitored. Reference data is used for the techno-economic assessment (cf., Tables A12 and A13). No historical data is available for a price forecast.

Figure 3 outlines the general project for CRR1 and ERR2 from a material flow perspective. Geotechnical and mine planning considerations are conceptual. The low mineral content estimated in Reference [53] is adopted to estimate conservatively (cf., Table A11). A homogeneous deposit is assumed. The tailings are mined in a dredging operation (cf., Figure S10) and processed on site in the existing processing plant at a constant rate over a 10-year period, followed by a 1-year rehabilitation period. The products leave the system boundaries at the mineral processing plant's outlet where the reference point is set. The target minerals are extracted with a multi-stage froth flotation as specified by Roemer [60] (cf., Table A16) based on a sampling campaign on the lower pond [53]. A pure industrial mineral concentrate ($BaSO_4$), a mixed sulphide concentrate containing base metals (Cu, Pb, Zn) and high-technology metals (Co, Ga, In), and mixed residues are produced. Tailings, commodity, and residue masses are estimated as dry matter.

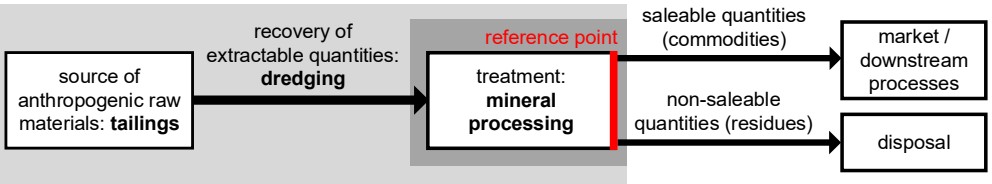

**Figure 3.** Tailings mining project Bollrich for the mineral RMs recovery scenarios (CRR1, ERR2) from a material flow perspective. The light grey and dark grey shaded fields illustrate the spatial and mineral processing system boundaries, respectively.

The database with fixed and variable parameters for the techno-economic assessment is given in Tables A14–A16. Energy flows are considered for tailings recovery and processing. Initial and intermediate investment costs for mining and processing equipment, and infrastructure, are included in the capital expenditure (CAPEX). Variable costs for mining, processing, electric and mechanical maintenance, administration, and general services are included in the operating expenditure (OPEX). Revenues are realised immediately. In ERR2, the mixed residues are sold to a recycling company for an application in construction materials. Mine site preparation costs are estimated to be low due to the simple mine plan, good mine site accessibility by road, and the availability of buildings for the processing plant and the operation's administration. Mine site rehabilitation costs such as for revegetation and environmental monitoring are considered. Assets and machinery are liquidated at the operation's end at a residual value of 10%.

Certain relevant aspects are out of the scope of this study: costs for preventing emissions during development, mining, transport and processing, for renewing the railway access, for removing roads and railway at mine closure, for treating and disposing of water from mining and processing, and downstream processing.

The uncertainty analysis comprises 3 price forecasts: pessimistic (p), mean (m), and optimistic (o), after which the respective scenarios are named (CRR1p, CRR1m, etc.). The pessimistic and optimistic forecasts refer to the lower and upper limits of the 95% confidence interval, respectively. $CuFeS_2$, PbS, and ZnS concentrate prices are estimated [62]. Prices for selling and costs for disposing of residues are fixed due to a lack of data. The mean price forecast (m), representing the most realistic case, is focussed. Material flow uncertainties are neglected as the dependence on price and cost variations is focussed.

### 3.2. Case Study Assessment

3.2.1. Environmental Assessment: Status Quo Risks

The area around the TSF is contaminated with heavy metals such as As, Cd, and Pb, which partially exceed the concentration threshold values for soil in parks and recreational areas in Germany [63,64]. However, the source of pollution could also be the former

transport of ores via the Bollrich area to smelters in Oker [65]. Hence, the TSF's contribution to the pollution is unknown.

No data is available on the TSF's impact on human health, local flora and fauna, and surface and groundwater as there currently is no monitoring in place [53]. Dust emissions from the TSF can be excluded due to the wet tailings storage. The neutralisation sludge is unlikely to emit dust as it hardens when being exposed to air [54]. Heavy-metal-laden seepage is collected at the foot of the dam and returned into the TSF [53]. However, the unsealed TSF base constitutes a risk for the release of contaminants [53]. A general safety concern is that the TSF is freely accessible (observed on Google Earth [50]), and there are several trails around the TSF (https://regio.outdooractive.com/oar-goslar/de/touren/#filter=r-fullyTranslatedLangus-,sb-sortedBy-0&zc=15,10.46323,51.90085, accessed on 16 January 2021). Hence, people who are not familiar with the area may come in direct contact with the TSF.

The main dam's stability in its current state and in the case of extreme rainfalls could be confirmed by conservative calculations [66]. However, 2 sinkholes in karstified zones in near vicinity to the TSF were reported [53]. The knowledge on the karstified zones is limited [53] so that the long-term risk for the TSF's stability is currently unknown.

### 3.2.2. Social Assessment: Stakeholder Considerations

The Harz region has an ore mining history ranging from the Middle Ages to the 1980s [52]. Today, the region is facing the challenges of demographic change, young people's emigration, a weak economy, and environmental burdens from former mining [52,65]. A particularity is the Goslar community's and city administration's strong awareness of the region's mining history, which is regarded as a cultural heritage and an important factor for tourism [52,65]. This can be observed in public social media such as the Goslar Tales forum: the category *Mines and Smelters* has 70 topics from 2011 to 2019 with 925 contributions (http://www.goslarer-geschichten.de/forum.php, accessed on 26 September 2020). The TSF's history, basic knowledge, opinions, and safety concerns on water quality are discussed, and photos and videos are shared.

The results of Bleicher et al. [52] are summarised: generally, RMs recovery from mine waste is regarded as a development opportunity for the Harz region, and the trust in scientists and the industry is shared by public media. Scientific institutions and the industry are identified as the current regional drivers of CRMs recovery from mine waste. All interviewed stakeholders were in favour of developing knowledge and technologies for mine waste valorisation, with the exception of minor criticism from an environmental activist about the presumption of scientists that good ideas are approved by everyone. However, environmental NGOs see RMs recovery from mine waste as an opportunity to at least partially rehabilitate the environment. The city's administration is interested in RMs recovery from mine waste since the establishment of a recycling industry might attract highly skilled workers, and the possible knowledge transfer with scientific institutions and the opportunity to test novel technologies is seen as one of the region's strengths.

### 3.2.3. Techno-Economic Assessment: Material Flow Analysis

No material flow takes place in NRR0 due to in-situ stabilisation. Figure 4 depicts the specific material flows for the RMs recovery scenarios (CRR1, ERR2) (cf., Figure A1 for a detailed production breakdown). Over a 10-year period, 7.1 million t of tailings are mined and processed. In CRR1, 2.7 million t of commodities (i.e., 38 wt% of total tailings), and 4.4 million t of mixed mineral residues are produced. The commodities consist of an industrial mineral and a mixed sulphide concentrate. In ERR2, all tailings are valorised. The commodities (CRR1, ERR2) leave the system boundaries for off-site conditioning.

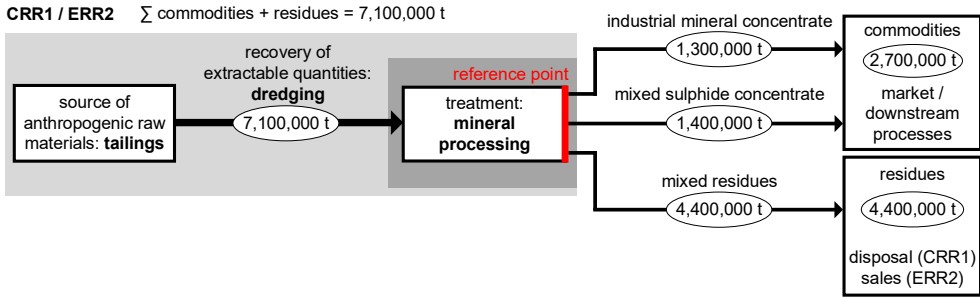

**Figure 4.** Material flow systems and 5-year material flows for the mineral RMs recovery scenarios (CRR1, ERR2). The light grey and dark grey shaded fields illustrate the spatial and mineral processing system boundaries, respectively. All figures were rounded to the sixth digit.

### 3.2.4. Techno-Economic Assessment: Discounted Cash Flow Analysis

Table 3 summarises the results of the DCF analysis (cf., Figures S15–S17). Generally, mineral RMs recovery is economically viable (CRR1m, ERR2m) under the project's current state of assessment. The DCF analysis yields positive NPVs in ERR2 regardless of the price forecast. The NPV in CRR1 becomes negative in the pessimistic forecast (CRR1p). The NPVs of NRR0, CRR1m, and ERR2m are EUR −124.5 million, EUR 73.9 million, and EUR 172.5 million, respectively. 98% of all costs in the rehabilitation scenario (NRR0) are attributed to the 5-year closure and leachate phase. In the mineral RMs recovery scenarios (CRR1m, ERR2m), the largest share of revenues is attributed to $BaSO_4$ with a 49% and 47% contribution, respectively, and a share of the total commodity masses of 64.4 wt% and 24.5 wt%, respectively. The second highest revenues are attributed to Zn with a contribution of 27% and 25%, respectively, and a ZnS share of the total commodity masses of 5.5 wt% and 2.1 wt%, respectively. The high-technology metals Co, Ga, and In contribute least to the revenues from RMs sales with a combined share of ca. 2% of total revenues and a combined share of total commodity mass of 0.6% and 0.02%, respectively.

**Table 3.** Results of the DCF analysis. The rehabilitation scenario (NRR0) has a project duration of 35 years. The RMs recovery scenarios (CRR1, ERR2) has a project duration of 11 years. The left column shows cost and revenue factors of the NPVs. Figures are given in millions of EUR.

| | Scenarios [1] | | | | | | |
|---|---|---|---|---|---|---|---|
| | **NRR0** | **CRR1p** | **ERR2p** | **CRR1m** | **ERR2m** | **CRR1o** | **ERR2o** |
| **NPV Factor** | | | | | | | |
| total NPV | −124.6 | −16.6 | 82.0 | 73.9 | 172.5 | 164.4 | 263.1 |
| *costs* | | | | | | | |
| CAPEX | - | −14.6 | −14.6 | −14.6 | −14.6 | −14.6 | −14.6 |
| OPEX | - | −29.1 | −29.1 | −29.1 | −29.1 | −29.1 | −29.1 |
| diesel | - | −3.4 | −3.4 | −5.1 | −5.1 | −6.9 | −6.9 |
| electric energy | - | −1.2 | −1.2 | −1.2 | −1.2 | −1.2 | −1.2 |
| residue disposal | - | −87.7 | - | −87.7 | - | −87.7 | - |
| rehabilitation | - | −4.0 | −4.0 | −4.0 | −4.0 | −4.0 | −4.0 |
| closure & leachate phase | −122.0 | - | - | - | - | - | - |
| aftercare phase | −2.6 | - | - | - | - | - | - |
| *revenues* | | | | | | | |
| $BaSO_4$ | - | 92.1 | 92.1 | 106.2 | 106.2 | 120.4 | 120.4 |
| Cu | - | 9.4 | 9.4 | 14.9 | 14.9 | 20.3 | 20.3 |
| Pb | - | 14.1 | 14.1 | 30.5 | 30.5 | 47.0 | 47.0 |
| Zn | - | 6.1 | 6.1 | 58.2 | 58.2 | 110.2 | 110.2 |
| Co | - | 0.7 | 0.7 | 2.6 | 2.6 | 4.6 | 4.6 |
| Ga | - | 0.3 | 0.3 | 0.7 | 0.7 | 1.0 | 1.0 |
| In | - | 0.2 | 0.2 | 2.1 | 2.1 | 4.1 | 4.1 |
| asset liquidation | - | 0.1 | 0.1 | 0.1 | 0.1 | 0.1 | 0.1 |
| residue sales | - | - | 11.0 | - | 10.9 | - | 10.9 |

[1] p: pessimistic price forecast (lower limit of 95% confidence interval), m: mean price forecast, o: optimistic price forecast (upper limit of 95% confidence interval).

Residue disposal is the highest cost factor in CRR1m with a share of 62% of total costs. The OPEX is the second highest cost factor in CRR1m and the highest in ERR2m with a share of total costs of 21% and 58%, respectively. In both scenarios, the smallest cost factor is electric energy consumption with a share of 0.8% and 2.4%, respectively.

### 3.2.5. Techno-Economic Assessment: Sensitivity and Uncertainty Analysis

The NPV is most sensitive to $BaSO_4$ price variations (cf., Figures A2 and A3). In CRR1m and ERR2m, a decreased $BaSO_4$ price by 69% and 100% yields an NPV decrease of 100% and 62%, respectively. In CRR1m, decreased Pb and Zn prices by 100% yields an NPV decrease of 42% and 79%, respectively. In ERR2m, a decreased Zn price by 100% yields an NPV decrease of 34%. The NPV is relatively insensitive to other price variations.

Residue disposal was the most influential cost factor in CRR1m, with a price increase of 84% yielding an NPV of zero. CAPEX and OPEX increases of 504% and 253% (CRR1m), respectively, and 1178% and 592% (ERR2m), respectively, yields NPVs of zero.

### 3.2.6. Legal Assessment: Basic Considerations

The legal aspects for a possible project execution have not been considered so far. The TSF is still monitored under Mining Law (State Office for Mining Energy and Geology (LBEG), personal communication, 16 September 2020). As for the right of mining, it needs to be assessed if the mining or waste legislation applies [67]. Goldmann et al. [53] rate the legal aspects for environmental protection as follows: strict legal restrictions and high efforts to achieve legal consent are expected since heterogeneous and high-quality flora and fauna ecosystems were identified during preliminary on-site inspections. It is likely that an environmental impact study and a concept to protect the ecosystems and/or to remediate impacts upfront are necessary. Potential impacts on the surrounding protected natural areas and landscapes need to be assessed. As for water protection, potential impacts on the river Gelmke in near vicinity (cf., Figure 2) and the nearby Ammentalbach need to be assessed. Potential impacts on groundwater are unclarified.

## 4. Discussion

### 4.1. Interpretation of the Case Study Results

The rating results are summarised in the categorisation matrix in Tables 4 and 5. The justification for the rating is given in Tables A17–A26. As no RMs are recovered in the rehabilitation scenario (NRR0), only the overall project is rated. The lowest rating in a category is chosen for the rating of the overall category (cf., Reference [68] (p. 37)).

For NRR0, the categorisation matrix shows that the knowledge on the TSF's geology has medium confidence (G2). The rehabilitation scenario's state of technological development has a low overall rating (F3) due to the uncertainty regarding possible ordnance, the conceptual operational design, the unclarified usability of TSF water, and the unclarified long-term storage safety. The infrastructural conditions (F1–F2) and rehabilitation planning (F2) are rated high. As only costs are incurred and as there currently is no knowledge on a potential financial support, the economics are rated low (E3.3a). As for the environmental aspects, the unclarified potential dust emission and in-situ cementation of reactive material lead to a low rating (E3.3b). As for the social aspects, only the retained landscape is rated positively (E2c). The legal aspects are generally underdeveloped (E3.3d).

In CRR1m and ERR2m, the project can be expected to be economically viable (E3.1a). However, the NPV in the pessimistic forecast for CRR1 is negative. ERR2 is more resilient in this respect due to the sales of the new residues. The favourable economics of ERR2 are highlighted in the overall category rating (E.3.1a) as opposed to CRR1 (E3.3a) due to the higher uncertainty in the pessimistic price forecast. The driving revenue factor is the $BaSO_4$ sales due to its relatively high grade (24.5 wt%), its high price compared to the other commodities, its high recovery rate (74%), and the forecasted price increase. The $BaSO_4$ price is relatively stable, with the largest price drop being ca. 17% in the past 20 years (cf., Figure S3). CRR1m is relatively insensitive to $BaSO_4$ price variations with the NPV

becoming negative at a decreased $BaSO_4$ price by 69%. ERR2 is more resilient with a $BaSO_4$ price drop to EUR 0, leading to a decreased NPV of 38%. In general, the presence of real estate, transportation, and utilities infrastructure reduces the mine development costs.

Residue disposal is the greatest cost factor in CRR1 with 64% of all costs, and it is the greatest economic risk with a price increase of 93% leading to a negative NPV. A price increase is possible if a further conditioning is necessary to meet the criteria of disposal sites. Regarding CAPEX and OPEX, CRR1m and ERR2m are relatively insensitive to cost variations, and they are regarded as economically viable given that the estimates are in the accuracy and contingency range for scoping studies of 50% and 30%, respectively [45].

For the upper pond, there is high uncertainty regarding geological knowledge on the neutralisation sludge, as well as the Co, Ca, and In contents (G3). The TSF's volume, and the $BaSO_4$ and base metal contents are well known (G2). Metallurgical testwork on the tailings from the upper pond is missing (F3), and it is unknown if the neutralisation sludge could be valorised in ERR2. These tailings might be difficult to process due to the high sulphate ion content [54]. If they need to be disposed of too, the disposal costs would increase in both scenarios (CRR1, ERR2). RMs recovery has a higher rating regarding environmental aspects as compared to rehabilitation only (NRR0). However, planning considerations such as the resettlement of rare flora and fauna still requires fundamental work (E3.3d), and the RMs efficiency (E3.3c) and preservation of RMs for future generations (E3.2c) in CRR1 could be improved. In contrast, the complete tailings valorisation (E1c) and high RM efficiency (E3.1c) are positively highlighted in the categorisation matrix. The development status of social aspects is generally low, just as for legal aspects (E3.3d).

For the individual RMs, a clear distinction in the geological and technological categories between the development status for $BaSO_4$ (G2F2), base metals (G2F2), $FeS_2$ (G2F1), and inert material (G2F1) can be seen as compared to the high-technology metals (G3F3). The development status for economic and environmental aspects is heterogeneous. Most RMs have a high economic importance or are CRMs in the EU, and all except for $FeS_2$ and inert material have a clear demand. The mean RM price forecast yields increasing $BaSO_4$, Co, and In prices (E3.1a); stagnant Pb and Zn prices (E3.2a); and decreasing Cu and Ga prices (E3.3a). For the new residues, the Pb solid matter content and dissolved Pb in leachate impede a disposal as inert waste (DK 0 class) (E3.2b) [61]. On the extreme ends, Ga and $FeS_2$ has the lowest (G3F3E3.3a) and highest (G2F1E3.2a) rating, respectively.

In sum, all 3 scenarios are rated equally in the overall rating in terms of the degree of confidence in the geological estimates and technical feasibility (G2F3). The scenarios differ in the economic performance with rehabilitation incurring costs only, and CRR1 having a higher uncertainty as compared to ERR2. Considering the proposed differentiation of the E category, the scenarios are categorised as G2/F3/E3.3a/E3.3b/E3.3c/E3.3d (NRR0), G2/F3/E3.3a/E3.2b/E3.3c/E3.3d (CRR1), and G2/F3/E3.1a/E3.2b/E3.3c/E3.3d (ERR2). The conversion into the current official UNFC categorisation yields G2F3E3 for all 3 scenarios. There is currently no class for this categorisation [44]. In comparison to the categorisation of G4F3E3 in the preceding screening study [43], only the G category could be improved.

**Table 4.** Categorisation matrix for the overall project rating of the rehabilitation scenario (NRR0) and the mineral RMs recovery scenarios (CRR1, ERR2).

| Factor | Scenario | | |
|---|---|---|---|
| | **NRR0** | **CRR1** | **ERR2** |
| | **UNFC G Category** | | |
| *geological conditions (relevant for project development)* | | | |
| (1) quantity | G2 | G2 | G2 |
| (2) quality | G2 | G2 | G2 |
| (3) homogeneity | G2 | G2 | G2 |
| | **UNFC F Category** | | |
| *TSF condition & risks (relevant for project development)* | | | |
| (4) ordnance | F3 | F3 | F3 |
| *mine planning considerations (relevant for project execution)* | | | |
| (5) mine/operational design | F3 | F3 | F3 |
| (6) metallurgical testwork | - | F3 | F3 |
| (7) water consumption | F3 | F1 | F1 |
| *infrastructure (relevant for project development)* | | | |
| (8) real estate | F1 | F1 | F1 |
| (9) mining & processing | - | F3 | F3 |
| (10) utilities | F2 | F2 | F2 |
| (11) transportation & access | F2 | F2 | F2 |
| *post-mining state (relevant for future impacts)* | | | |
| (12) residue storage safety | F3 | F3 | F3 |
| (13) rehabilitation | F2 | F2 | F2 |
| | **UNFC E Category** [1] | | |
| *microeconomic aspects (relevant for project development)* | | | |
| (14) economic viability | E3.3a | E3.1a | E3.1a |
| (15) economic uncertainty | - | E3.3a | E3.1a |
| *financial aspects (relevant for project development)* | | | |
| (16) investment conditions | - | E3.1a | E3.1a |
| (17) financial support | E3.3a | E3.1a | E3.1a |
| *environmental impacts during project execution* | | | |
| (18) air emission | E3.3b | E3.1b | E3.1b |
| (19) liquid effluent emission | E3.1b | E3.1b | E3.1b |
| (20) noise emission | E3.2b | E3.2b | E3.2b |
| *environmental impacts after project execution* | | | |
| (21) biodiversity | E3b | E3b | E3b |
| (22) land use | E3.2b | E3.2b | E3.2b |
| (23) material reactivity | E3.3b | E3.1b | E3.1b |
| *social impacts during project execution* | | | |
| (24) local community | E3.3c | E3.2c | E3.2c |
| (25) health & safety | E3.3c | E3.3c | E3.3c |
| (26) human rights & business ethics | E3.3c | E3.3c | E3.3c |
| *social impacts due to project execution* | | | |
| (27) wealth distribution | E3.3c | E3.3c | E3.3c |
| (28) investment in local human capital | E3.3c | E3.3c | E3.3c |
| (29) degree of RM recovery | E3.3c | E3.2c | E1c |
| (30) RM valorisation | E3.3c | E3.3c | E3.1c |
| *social impacts after project execution* | | | |
| (31) aftercare | E3c | E1c | E1c |
| (32) landscape | E2c | E1c | E1c |
| *legal situation (relevant for project development)* | | | |
| (33) right of mining | E3.3d | E3.3d | E3.3d |
| (34) environmental protection | E3.3d | E3.3d | E3.3d |
| (35) water protection | E3.3d | E3.3d | E3.3d |
| **total rating** | G2 | G2 | G2 |
| | F3 | F3 | F3 |
| | E3.3a | E3.3a | E3.1a |
| | E3.3b | E3.2b | E3.2b |
| | E3.3c | E3.3c | E3.3c |
| | E3.3d | E3.3d | E3.3d |

[1] a: economic aspects, b: environmental aspects, c: social aspects, d: legal aspects.

**Table 5.** Categorisation matrix for the subproject rating for individual RMs (CRR1, ERR2).

| Factor | BaSO$_4$ | Cu | Pb | Zn | Co | Ga | In | FeS$_2$ | Inert Material [1] |
|---|---|---|---|---|---|---|---|---|---|
| | | | | | **Subprojects for RMs** | | | | |
| | | | | **UNFC G Category** | | | | | |
| *geological conditions (relevant for project development)* | | | | | | | | | |
| (36) quantity | G2 | G2 | G2 | G2 | G3 | G3 | G3 | G2 | G2 |
| (37) quality | G2 | G2 | G2 | G2 | G3 | G3 | G3 | G2 | G2 |
| (38) homogeneity | G2 | G2 | G2 | G2 | G3 | G3 | G3 | G2 | G2 |
| | | | | **UNFC F Category** | | | | | |
| *mine planning considerations (relevant for project execution)* | | | | | | | | | |
| (39) recoverability | F2 | F2 | F2 | F2 | F3 | F3 | F3 | F1 | F1 |
| | | | | **UNFC E Category [2]** | | | | | |
| *microeconomic aspects (relevant for project development)* | | | | | | | | | |
| (40) demand | E3.1a | E3.1a | E3.1a | E3.1a | E3.1a | E3.1a | E3.1a | E3.2a | E3.3a |
| (41) RM criticality | E1a | E2a | E2a | E2a | E1a | E1a | E1a | E2a | E3a |
| (42) price development | E3.1a | E3.3a | E3.2a | E3.2a | E3.1a | E3.3a | E3.1a | - | - |
| *impacts after project execution* | | | | | | | | | |
| (43) solid matter | - | E3.1b | E3.2b | E3.1b | - | - | - | - | E1b |
| (44) eluate | E3.1b | E3.1b | E3.2b | E3.1b | - | - | - | - | E1b |
| **total rating** | G2 | G2 | G2 | G2 | G3 | G3 | G3 | G2 | G2 |
| | F2 | F2 | F2 | F2 | F3 | F3 | F3 | F1 | F1 |
| | E3.1a | E3.3a | E3.2a | E3.2a | E3.1a | E3.3a | E3.1a | E3.2a | E3.3a |
| | E3.1b | E3.1b | E3.2b | E3.1b | - | - | - | - | E1b |

[1] Wissenbach shales & ankerit. [2] a: economic aspects, b: environmental aspects, c: social aspects, d: legal aspects.

### 4.2. Reconciliation of Stakeholder Perspectives with an Application of the UNFC Principles

Environmental NGOs' perspective: the TSF Bollrich constitutes an ecological burden in a sensitive environment with high potential long-term environmental and social risks [43]. Indeed, the TSF's current geomechanical state is stable, but it requires constant maintenance such as the removal of large trees and assuring seepage in the main dam [66]. The TSF is an upstream dam type, which is the most vulnerable type [16,20]. The lacking knowledge on the karstified zones in the area and the former occurrence of sinkholes near the TSF are currently rated as non-problematic [53]. However, for a conservative approach, the risk must be rated high due to the uncertainty. A sudden release of the contained masses and toxic elements would cause widespread environmental destruction and social issues, and would threaten human lives [43]. Therefore, the long-term physical and chemical risks and associated legacy costs are regarded as a necessity to act. Hence, early actions are preferable, and the rehabilitation costs (NRR0) can be seen as external costs borne by society to prevent harm. As the TSF is integrated well into the landscape, being visible only from nearby hills or from close up, the benefit of NRR0 is that the current landscape is mostly retained. On top, NRR0 has a relatively short duration of perceptible works on the TSF of 5 years. Hence, negative environmental and social impacts due to project execution are kept at a minimum as compared to RMs recovery (CRR1, ERR2). However, stabilising the tailings impedes a future RMs recovery. On top, rehabilitation incurs costs only so that a combination with RMs recovery (CRR1, ERR2) is preferable. Since the new residues in CRR1 consume land due in a disposal site and since future emissions cannot be excluded as the storage conditions are currently unclear, ERR2 is preferable.

Private investors' perspective: TSF rehabilitation (NRR0) generates relatively high revenues. However, the TSF Bollrich is an economically viable source of important RMs. Since a domestic RMs recovery can contribute to reducing RM supply risks by diversifying the sourcing of CRMs on a national level, a private company could benefit from a positive public perception when engaging in RMs recovery. As CRR1 and ERR2 include environmental rehabilitation, they reduce the anthropogenic footprint. As the highest revenues of all scenarios are generated in ERR2, and as there is a certain economic risk in CRR1 shown with the pessimistic price forecast, ERR2 is preferable economically.

Goslar city administration's perspective: NRR0 is in line with the city development goals [65] by restoring the recreational qualities of the TSF area in a relatively short period.

However, the anthropogenic footprint is not reduced and the tailings' long-term stability is unclear [69] so that future measures might be necessary. With RMs recovery (CRR1, ERR2), the city administration saves rehabilitation expenses. An intensified interaction of industry and scientific institutions could strengthen the region in the long run. However, the short duration of active works (CRR1) thwart the goal to establish long-term high-quality jobs and to attract investors who seek long-term opportunities [65]. Such opportunities are created in ERR2 so that the Harz region's challenge of a weak economic structure and emigration of young people can be tackled [52], and an innovative recycling industry can be established [65]. Dealing with the region's environmental legacy from former mining is seen by the city administration of Goslar as a key challenge for a sustainable development [65] so that negative impacts of new residues must be avoided (ERR2).

Résumé: with the application of the UNFC-principles, the advantages and disadvantages of all 3 scenarios could be made visible for all 3 stakeholders. The overview of all factors shows that all 3 stakeholder interests are best fulfilled with the RMs recovery scenario ERR2 in which most benefits are generated, namely, environmental rehabilitation, economic revenues, and long-term regional development. In the assessed constellation, the city administration of Goslar would be a particularly eligible main project driver under compulsory consideration of the enablers environmental NGOs and private investors.

### 4.3. Path Forward for the Case Study Bollrich

For the RMs recovery scenarios (CRR1, ERR2), a higher rating of the project as *potentially viable* (G2F2E2) requires the following aspects to be addressed: the extent of karstified zones needs to be investigated to better assess the risk of a potential damage to the TSF. The amount of dam material, and the amount, composition, distribution and valorisability of neutralisation sludge need to be investigated. Furthermore, a solution is required for the discharge of the Rammelsberg mine water, preferably with a recovery of RMs such as Zn. The costs for residue disposal (CRR1) and conditioning for an application in construction materials (ERR2) needs to be investigated. To enhance RM efficiency, a potential concentrate buyer needs to be willing to valorise the $FeS_2$ and to recover the high-technology metals. It should be investigated if all residues in ERR2 can be valorised. The recoverability of As, Cd, Cr, Ni, and Tl needs to be investigated as they are important in high-technology applications, e.g., robotics or decarbonised energy production [70].

A milestone is the determination of site-specific processing costs for which reference values are used in this article. An economic estimation after taxes and other governmental charges are required to make it comparable across country borders [71]. An uncertainty analysis on tailings mass could account for errors in the geological estimates.

In terms of legal aspects, fundamental work must be carried out such as the estimation of costs and the duration of clarifying legal barriers, the engagement of authorities, and the drafting of applications. As for environmental aspects, the present flora and fauna needs to be inventoried in detail; measures for the compensation of environmental impacts need to be drafted; and rehabilitation, environmental monitoring, and post-closure land use plans need to be conceptualised. For the endorsement of a project plan, a disposal site for residues needs to be determined, and a transportation concept must be developed.

A comprehensive systematic stakeholder assessment is required. The process should be transparent and clearly structured to enable a fact-based discussion at all times. For all scenarios, the TSF's long-term risks need to be weighed against the temporary disturbance of local nature and communities, potential long-term regional benefits such as environmental rehabilitation, and the local recruitment of workforce.

### 4.4. Integrating Sustainability Aspects into Raw Materials Classification

RMs recovery from tailings can have certain benefits: processing the already ground tailings is less energy-intense than processing ores under similar conditions [72]. The potential savings are high since ore crushing and grinding are the most energy-intense processes with ca. 40% of a mine's energy consumption [73,74]. Moreover, it is increasingly

acknowledged that aspects other than the RMs have to be considered in present-day RMs assessments [52]. RMs recovery from tailings offers the opportunity to rehabilitate the environment [12,75], which can reduce environmental and social risks. Hence, tailings can be regarded as a secondary RM source with a lower social conflict potential than ores [11].

The challenge is to identify and communicate these potential benefits, especially for environmental and social aspects [46]. Indeed, geological and techno-economic aspects can be assessed with established methods from the conventional CRIRSCO classification [45], but it is unsuitable for capturing sustainability aspects [43,49]. In contrast, the UNFC recognises environmental and social aspects as potential driving factors, integrating them into the classification [44]. Current shortcomings of the UNFC are its lacking practicability [8], user guidance [43,49], specification of knowledge which must be generated in very preliminary studies [49], and standardised assessment and classification template for anthropogenic RMs including key factors which must be considered [47,49]. This article demonstrates how one can be guided through a practical UNFC application. Established methods from the conventional mineral RMs classification are combined with methods to account for environmental and social benefits. With the following aspects, the developed approach supports the integration of sustainability aspects into RMs classification:

First, the report of on-site exploration data by Goldmann et al. [53] on the TSF Bollrich documents relevant aspects extensively but it lacks a frame for an overall rating. In their report, a techno-economic classification of the tailings in terms of conventional *resources* or *reserves* as well as the determination of cut-off grades was not possible due to the geological uncertainties [53]. Environmental and legal aspects are discussed separately, but they do not contribute to the classification. This is common in current classification practice, which focusses on economic aspects [16,40]. Therefore, current practice cannot fully reflect a project's potentials. In contrast, the presented UNFC-compliant assessment and classification approach provides a comprehensive framework to communicate the development status of the TSF Bollrich case study by considering all relevant geological, technological, and environmental-socio-economic aspects on site during exploration.

Second, mining companies worldwide are increasingly recognising that their economic interests need to be aligned with social values for long-term success [6,23,76]. However, the reinterpretation of waste as a RM source requires a change of mindset [52]. In this context, a challenge is to create a common understanding of sustainable acting as local stakeholders' perspectives on sustainable mining often diverge [77]. Hence, the sustainable prospects of a potential project need to be communicated transparently to local communities in the project development phase to create a common understanding. Thus, the developed assessment and classification approach offers the opportunity to integrate a stakeholder assessment in the decision-making process. The needs of local stakeholders are particularly addressed in terms of impacts related to land use, the environment, and health.

Third, the example of the Harz region highlights the importance of including social aspects such as involving local communities in the development of RMs recovery projects and transparently communicating potential long-term impacts on former contaminated sites: although the Mansfeld area is comparable to the Goslar area, the local population is sceptical about RMs recovery due to dishonest communication and selfish behaviour of potential project developers in the past [52]. Especially in densely populated areas, social conflicts can arise. The inclusion of local values, such as those expressed by the town council as the elected representative of local citizens, can help to improve the sustainability of a project and influence a project assessment in terms of enhancing the common good [77].

Fourth, the developed categorisation matrix addresses several issues: in the classification of tailings with conventional practice, the RM potential beside the target RM potential is usually not captured, e.g., References [37–39]. This means that part of the RM potential remains unassessed. The distinct classification of the individual RMs in the categorisation matrix highlights the potentials of and barriers to their recovery. The heat map-like visualisation of the categorisation enables a quick comparison of all aspects with each other, promoting a transparent communication of the assessment results. For instance, in each of

the scenarios, the impairment of local ecosystems around the TSF Bollrich are captured in the categorisation matrix. Consequently, a project developer is required to comment on how further measures can be taken to overcome the scenario-specific barriers. As another example, even a longer duration of the RMs recovery scenarios (CRR1, ERR2) could be considered more favourable than the relatively short impairment caused by the rehabilitation scenario (NRR0) due to the long-term benefits resulting from the risk reduction associated with the removal of the tailings. In a stakeholder assessment, all relevant stakeholders can question the factors considered in order to reach a mutually agreed decision. In the course of the study, consensus building can be documented and evaluated.

Fifth, the case study shows how the application of the UNFC principles can reconcile 3 different stakeholder perspectives: the TSF-related long-term risks are identified as the main project drivers. Considering the remediation costs as external costs borne by society enables a comparison of the monetary impacts of the TSF in case of rehabilitation (NRR0) with those of the other scenarios (CRR1, ERR2). Scrutinising the considered stakeholder perspectives leads to the following common values: minimisation of physico-chemical risks associated with the TSF, minimisation of emissions to the environment during any operation, achievement of a long-term aftercare-free state after project execution, and the preservation of the area's recreational value and ecosystem quality. On this basis, the RMs recovery scenario ERR2 should be prioritised since it addresses all common values.

*4.5. Development Potential of the Assessment and Classification Approach*

A comparison of the classification result from the screening of the TSF Bollrich (G4F3E3) in Reference [43] to the result from this article (G2F3E3) shows that the improvements in the E and F categories are not reflected in the overall rating. This can be explained with the selected factors and indicators to measure the development status, especially for the social and legal aspects. A comparison of the factors and indicators applied in this study with other case studies could show if they all suit the scope of a very preliminary study or if some of them should be applied in more developed studies. Additionally, the low rating in the E and F categories can be explained with the procedure to choose the lowest rating in a category as the overall rating. An example is the rating of economic aspects for the RM Cu: despite the favourable rating of the demand (E3.1a) and RM criticality (E2a), the low rating of the forecasted decreasing price development (E3.3a) is determinant. This issue could be resolved by weighting factors for instance. It is worth noting that there is currently no class defined for a rating as G2F3E3. A proposal is made for a possible description: *based on very preliminary results, a prospective project has been identified as a potential source of RMs for which further studies are required to justify further development.*

Factors related to the impact on global warming are not considered in this study. This could be remediated by performing a life cycle assessment (LCA). It enables the consideration of external costs, and it was also used in conjunction with the UNFC [78]. Another advantage is that it allows for a comparison to projects from primary mining [78]. Regarding tailings, the LCA has been used to assess aspects such as environmental impacts in early phases of mine planning [79], and TSF site management and closure scenarios [80]. For RMs recovery from tailings, an LCA should provide decision-makers with information on environmental impacts which could be compared with primary mining. In general, the LCA requires site-specific data for a detailed analysis of processes and their impacts [81]. The LCA performed by Goldmann et al. [53] for the conceptualised dredging system shows that an LCA in very preliminary studies can be applied to assess different mining options. The use of LCAs in early project development phases on aspects such as mineral processing and a possible contribution to the classification must yet be examined.

## 5. Conclusions and Recommendations

To recapitulate, the deposition of tailings in TSFs impacts the environment and local communities and can even threaten human health [16]. These impacts could be aggravated

in the future due to a climate-change-induced increased likelihood of extreme weather occurrences [20]. At the same time, the global tailings production is increasing due to an increasing demand for highly important RMs, which are forecasted to at least double between 2010–2050 [4,5]. The increasing RM demand could partially be met by using the RM potential of tailings: 10–20% of all technospheric metal RMs are estimated to be deposited in landfills and TSFs; metal grades in tailings can be as high as in ores [40]. Technological advancements enable the exploitation of the residual metals content [29,82] or the valorisation in construction materials [83,84]. RMs recovery from tailings can also be an opportunity to reduce the environmental and social impacts of TSFs [75]. For the re-interpretation of tailings as a source of RMs, the potential benefits of and barriers to their exploitation need to be captured and assessed holistically. The assessment shows that the TSF Bollrich is an economically interesting source of $BaSO_4$; the base metals Cu, Pb, and Zn; and the high-technology metals Co, Ga, and In. Removing the TSF has positive long-term environmental impacts. However, there is high uncertainty regarding geological knowledge and technological extractability of the CRMs. An issue is that the applied social and legal factors are generally underdeveloped.

The research questions are answered: (1) the tailings deposit Bollrich is an example of a RMs recovery project which takes place in a complex environment where the influence of various site-specific stakeholders needs to be considered. With a UNFC-compliant approach, different stakeholder perspectives can be addressed in order to derive a commonly acceptable solution. In the case study, the enhanced mineral RMs recovery scenario ERR2 aligns the interests of environmental NGOs, private investors, and the city administration of Goslar: environmental rehabilitation to protect the TSF's vulnerable environment, the generation of profits, and a long-term regional development. It can therefore be concluded that a UNFC-compliant assessment is suitable for identifying areas of conflict between economic, environmental and social interests, and for achieving a generally acceptable solution. (2) It is suggested that for very preliminary studies, aspects relevant for project development and execution, impacts due to project execution, and impacts after project execution should be considered. Furthermore, the availability of primary on-site exploration data and secondary research data could be regarded as a prerequisite for a very preliminary study on tailings. As tailings usually contain multiple RMs, a comprehensive overview of the RM potential with differentiation of individual RMs is required. The data must allow for an initial assessment of the following aspects: (i) characterisation and quantification of the total and individual RM content, (ii) laboratory investigation of processability, (iii) technological conceptualisation of project execution and aftercare measures, (iv) DCF analysis, (v) inventory on present rare flora and fauna, (vi) status quo environmental risk assessment, and (vii) identification of relevant stakeholders. After a clarification of these aspects, a project can be advanced to a *preliminary study*. (3) The identification and communication of sustainability aspects in RMs classification poses a challenge. Despite a project's impact on its local environment and communities, related site-specific project potentials and barriers are usually not considered. The example of the Harz region demonstrates that, in addition to conventional economic interests, a site-specific approach is essential from the beginning of project development. The example of the tailings deposit Bollrich shows that an integration of local sustainability aspects into the assessment, represented by the development goals of the city administration of Goslar, can give a strong impulse for project development: strengthening the regional industrial role, creating high-value jobs, and developing tourism. The developed UNFC-compliant categorisation matrix captures the development status of specified factors and communicates the results in a quickly understandable manner in a heat-map-like style. Hence, it enables a point-by-point comparison of different scenarios so that the individual potentials and benefits become clear. In this way, the most auspicious option can be quickly identified, and its development can be justified.

Recommendations made: as for the case study TSF Bollrich, enhance the geological knowledge on the metalliferous CRMs; investigate the processability of the neutralisation

sludge; assess the recoverability of As, Cd, Cr, and Tl; and consider a direct valorisation of RMs in the Rammelsberg mine water. If the RMs recovery project is executed, the city administration's tax revenues could be used to rehabilitate other contaminated areas from former mining activities. In this way, the local community hosting the mining activity can benefit directly from it, which is uncommon in current practice [77]. Thus, RMs recovery from the TSF Bollrich could serve as a role model for a sustainable development of the Harz region. As for the developed approach, investigate if all selected factors and indicators, especially those for social and legal aspects, are suitable for very preliminary studies. Correspondingly, determine which factors are necessary and which are optional in very preliminary studies. Since the overall rating does not properly reflect the improvements made and deficits encountered in the course of several studies, introduce a reporting to support decision-making. As for the development of an anthropogenic RMs management, a database for the assessment of the global anthropogenic RM potential needs to be established. For this, waste producers could be obligated by law to report on all contained RMs in their wastes. Lastly, UNFC-compliant case studies on anthropogenic RMs are currently very labour-intensive due to a lack of experience. More UNFC-compliant case studies are needed to derive a reference base of project potentials and barriers. This would provide future studies with a benchmark for a quick recognition of a project's prospects of reaching the next level of maturity.

**Supplementary Materials:** Figure S1: Results of autoregressive electric energy price forecast based on yearly historical data from 2014 to 2020 from Statista [85]. The blue line on the right-hand side depicts the mean price forecast, and the blue and grey areas represent the 95% and 75% confidence intervals, respectively, Figure S2: Results of autoregressive diesel price forecast based on yearly historical data from 1950 to 2020 from Statista [86]. The blue line on the right-hand side depicts the mean price forecast, and the blue and grey areas represent the 95% and 75% confidence intervals, respectively, Figure S3: Results of autoregressive $BaSO_4$ price forecast based on yearly historical data from 2011 to 2020 from the USGS [87–90]. The blue line on the right-hand side depicts the mean price forecast, and the blue and grey areas represent the 95% and 75% confidence intervals, respectively, Figure S4: Results of autoregressive Co price forecast based on yearly historical data from 1996 to 2020 from the USGS [87,89–93]. The blue line on the right-hand side depicts the mean price forecast, and the blue and grey areas represent the 95% and 75% confidence intervals, respectively, Figure S5: Results of autoregressive Cu price forecast based on monthly historical data from 1999 to 2021 from IndexMundi [94]. The blue line on the right-hand side depicts the mean price forecast, and the blue and grey areas represent the 95% and 75% confidence intervals, respectively, Figure S6: Results of autoregressive Ga price forecast based on yearly historical data from 1999 to 2020 from the USGS [87,89–93]. The blue line on the right-hand side depicts the mean price forecast, and the blue and grey areas represent the 95% and 75% confidence intervals, respectively, Figure S7: Results of autoregressive In price forecast based on yearly historical data from 1999 to 2020 from the USGS [87,89–93]. The blue line on the right-hand side depicts the mean price forecast, and the blue and grey areas represent the 95% and 75% confidence intervals, respectively, Figure S8: Results of autoregressive Pb price forecast based on monthly historical data from 1999 to 2021 from IndexMundi [95]. The blue line on the right-hand side depicts the mean price forecast, and the blue and grey areas represent the 95% and 75% confidence intervals, respectively, Figure S9: Results of autoregressive Zn price forecast based on monthly historical data from 1999 to 2021 from IndexMundi [96]. The blue line on the right-hand side depicts the mean price forecast, and the blue and grey areas represent the 95% and 75% confidence intervals, respectively, Figure S10: Conceptual mine plan and processing schematic. The light grey shaded field indicates the spatial system boundaries and the dark grey shaded fields indicate products (adapted after Goldmann et al. [53]), Figure S11: Results of the sensitivity analysis of the conventional mineral RMs recovery scenario (CRR1p) with pessimistic price forecast and a discount rate of 15%, Figure S12: Results of the sensitivity analysis of the conventional mineral RMs recovery scenario (CRR1o) with optimistic price forecast and a discount rate of 15%, Figure S13: Results of the sensitivity analysis of the enhanced mineral RMs recovery scenario (ERR2p) with pessimistic price forecast and a discount rate of 15%, Figure S14: Results of the sensitivity analysis of the enhanced mineral RMs recovery scenario (ERR2o) with optimistic price forecast and a discount rate of 15%, Figure S15: Comparison

of costs, revenues and NPVs for the mean price forecast of the 3 scenarios with no mineral RMs recovery (NRR0), conventional mineral RMs recovery (CRR1m) and enhanced mineral RMs recovery (ERR2m). With a discount rate of 15%, NRR0 is discounted over a period of 35 years, and CRR1m and ERR2m over a period of 11 years, Figure S16: Comparison of costs, revenues and NPVs for the pessimistic price forecast of the 3 scenarios with no mineral RMs recovery (NRR0), conventional mineral RMs recovery (CRR1p) and enhanced mineral RMs recovery (ERR2p). With a discount rate of 15%, NRR0 is discounted over a period of 35 years, and CRR1p and ERR2p over a period of 11 years, Figure S17: Comparison of costs, revenues and NPVs for the optimistic price forecast of the 3 scenarios with no mineral RMs recovery (NRR0), conventional mineral RMs recovery (CRR1o) and enhanced mineral RMs recovery (ERR2o). With a discount rate of 15%, NRR0 is discounted over a period of 35 years, and CRR1o.

**Author Contributions:** Conceptualisation, R.S.; methodology, R.S.; validation, R.S., S.H.-A.; resources, R.S.; writing—original draft preparation, R.S.; writing—review and editing, R.S., S.H.-A.; visualisation, R.S.; project administration, R.S.; funding acquisition, R.S., S.H.-A. All authors have read and agreed to the published version of the manuscript.

**Funding:** This research was funded by the German Ministry of Research and Education (BMBF) as part of the research project ADRIANA (Client II programme), grant agreement number 033R213A-D.

**Institutional Review Board Statement:** Not applicable.

**Informed Consent Statement:** Not applicable.

**Data Availability Statement:** This research used publicly available data available in the referenced sources. The database can be found in the Appendix A and supplementary materials.

**Acknowledgments:** The authors are thankful to Bernd G. Lottermoser for his comments and to Jonas Krampe for providing the R code. In addition, the authors would like to express their deep gratitude to two anonymous reviewers who helped to improve the manuscript.

**Conflicts of Interest:** The authors declare no conflict of interest. The funders had no role in the design of the study; in the collection, analyses, or interpretation of data; in the writing of the manuscript; or in the decision to publish the results.

## Abbreviations

| Abbreviation/Unit | Description |
| --- | --- |
| Ag | lat. *argentum* (silver) |
| Al | aluminium |
| Au | lat. *aurum* (gold) |
| $BaSO_4$ | barium sulphate (barite) |
| Cd | lat. *cadmia* (cadmium) |
| Co | cobalt |
| Cu | lat. *cuprum* (copper) |
| $CuFeS_2$ | copper iron disulphide (chalcopyrite) |
| Fe | lat. *ferrum* (iron) |
| $FeS_2$ | iron disulphide (pyrite) |
| Ga | lat. *gallia* (gallium) |
| In | indium |
| Mn | manganese |
| Mo | molybdenum |
| Ni | nickel |
| Pb | lat. *plumbum* (lead) |
| PbS | lead sulphide (galena) |
| Tl | lat. *tellus* (tellurium) |
| Zn | zinc |
| ZnS | zinc sulphide (sphalerite) |
| ADRIANA | Airborne spectral Detection of Reusable Industry mAterials in tailiNgs fAcilities |

| | | |
|---|---|---|
| BMBF | German Ministry of Research and Education | |
| CAPEX | capital expenditure | |
| CL:AIRE | Contaminated Land: Applications in Real Environments | |
| CRM | Critical Raw Material | |
| DCF | discounted cash flow | |
| E | East | |
| EC | European Commission | |
| EU | European Union | |
| LOM | Life of Mine | |
| N | North | |
| NPV | net present value | |
| OPEX | operating expenditure | |
| Qty. | quantity | |
| RM | raw material | |
| TSF | tailings storage facility | |
| UNECE | United Nations Economic Commission for Europe | |
| UNFC | United Nations Framework Classification for Resources | |
| UNFC E category | represents environmental-socio-economic viability | |
| UNFC F category | represents technical feasibility | |
| UNFC G category | represents degree of confidence in the geological estimate | |
| USGS | U.S. Geological Survey | |
| W | West | |
| °C | degree Celsius (unit of temperature on the Celsius scale) | |
| μm | micrometre (unit of length, equivalent to $10^{-6}$ metres) | |
| a | year | |
| km | kilometre (unit of length, equivalent to $10^3$ metres) | |
| kW | kilowatt (SI-derived unit of power) | |
| kWh | kilowatt-hour (SI-derived unit of energy) | |
| l | litre (SI-derived unit of volume, equivalent to $10^{-3}$ m$^3$) | |
| m | metre (SI unit of length) | |
| m$^2$ | square metre (SI-derived unit of surface) | |
| m$^3$ | cubic metre (SI-derived unit of volume) | |
| mm | millimetre (unit of length, equivalent to $10^{-3}$ metres) | |
| t | metric tonne (unit of weight, equivalent to 1000 kilograms) | |

## Appendix A

**Table A1.** Degree of confidence in the geological estimates (G) for the overall project rating with the UNFC-compliant categorisation matrix.

| Factor | Explanation | Dependence on | Modification after | Indicator & UNFC Rating |
|---|---|---|---|---|
| *Geological conditions (relevant for project development)* | | | | |
| (1) quantity | amount of target RMs | ore quality, former processing efficiency, deposit volume | [45] | degree of geological certainty: high (G1) / medium (G2) / low (G3) |
| (2) quality | physico-chemical properties of target RMs | former processing, storage conditions | [45] | degree of geological certainty: high (G1) / medium (G2) / low (G3) |
| (3) homogeneity | distribution of target RMs inside the deposit | manner of former deposition | [24] | degree of geological certainty: high (G1) / medium (G2) / low (G3) |

**Table A2.** Technical feasibility (F) for the overall project rating with the UNFC-compliant categorisation matrix.

| Factor | Explanation | Dependence on | Modification after | Indicator & UNFC Rating |
|---|---|---|---|---|
| *TSF condition & risks (relevant for project development)* | | | | |
| (4) ordnance | unexploded ordnance from armed conflicts | regional history, former searching activities | - | degree of knowledge:<br>non-existence proven (F1)<br>existence proven (F2)<br>unclarified (F3) |
| *Mine planning considerations (relevant for project execution)* | | | | |
| (5) mine/operational design | optimising RMs recovery under consideration of strategic goals & restrictions | geological knowledge on deposit, project planning phase, quality of model assumptions, legal restrictions | [45] | level of detail of planning:<br>extended (incl. detailed operational factors) (F1)<br>advanced (incl. pit configuration & processing scheme) (F2)<br>basic (conceptual) (F3) |
| (6) metallurgical testwork | investigation of possible methods for mineral processing | sampling techniques, representativeness of test feed, testing techniques | [45] | degree of research on mineral processability:<br>industrial scale (F1)<br>pilot scale (F2)<br>laboratory scale (F3) |
| (7) water consumption | demand of fresh water supply for mining & processing | available water resources, water efficiency of mining system | [13,97,98] | percentage of recycled water:<br>high (>80%) (F1)<br>medium (50–80%) (F2)<br>low (<50%) (F3) |
| *Infrastructure (relevant for project development)* | | | | |
| (8) real estate | availability of land & reusability of buildings | former mine closure, current land use, time lapsed after abandonment | [45] | condition of infrastructure:<br>highly developed (fully reusable) (F1)<br>acceptable (usable after upgrade) (F2)<br>bleak (requires (re-)construction) (F3) |
| (9) mining & processing | reusability of equipment related to general services, mining & processing | former mine closure, current land use, time lapsed after abandonment | [45] | condition of equipment:<br>highly developed (fully reusable) (F1)<br>acceptable (usable after upgrade) (F2)<br>bleak (requires new acquisition) (F3) |
| (10) utilities | access to utilities supply lines (e.g., electricity) | mine closure & time lapsed after abandonment, current land use, proximity to human settlements | [45] | condition of infrastructure:<br>highly developed (full access) (F1)<br>acceptable (access after upgrade) (F2)<br>bleak (requires (re-)construction) (F3) |
| (11) transportation & access | access to mine & markets via air, road, railway, or waterway | topography, former mine closure, current land use, time lapsed after mine abandonment, proximity to human settlements | [45] | condition of infrastructure:<br>highly developed (fully reusable) (F1)<br>acceptable (usable after upgrade) (F2)<br>bleak (requires (re-)construction) (F3) |
| *Post-mining state (relevant for future impacts)* | | | | |
| (12) residue storage safety | ability of new storage facility to safely store new residues for an indefinite time period | amount of new residues, topography, type of construction, climate, regional seismic activity | [13,98–100] | suitability of new disposal site for safe storage:<br>high degree of safety proven (F1)<br>preliminary assertion of safety (F2)<br>unsafe or unclarified (G3) |
| (13) rehabilitation | process of recontouring, revegetating, & restoring the water & land values | residue characteristics, local ecosystem, landscape, environmental laws, local climate | [101] | level of detail of planning:<br>concrete (F1)<br>conceptual (F2)<br>none (F3) |

**Table A3.** Economic viability (E a) for the overall project rating with the UNFC-compliant categorisation matrix.

| Factor | Explanation | Dependence on | Modification after | Indicator & UNFC Rating |
|---|---|---|---|---|
| *Microeconomic aspects (relevant for project development)* | | | | |
| (14) economic viability | economic returns from project | mine planning, RMs prices, costs of input factors (labour, energy, materials), payments to public sector (e.g., taxes) | [45,97] | discounted cash flow over projected LOM: positive (NPV >> 0€) (E3.1a) neutral (NPV~0€) (E3.2a) negative (NPV << 0€) (E3.3a) |
| (15) economic uncertainty | overall uncertainty of economic estimates | degree of detail in planning, data quality of economic estimate | [45] | uncertainty of cash flow in pessimistic scenario: low (NPV >> 0€) (E3.1a) medium (NPV~0€) (E3.2a) high (NPV << 0€) (E3.3a) |
| *Financial aspects (relevant for project development)* | | | | |
| (16) investment conditions | conditions concerning taxes, royalties, & other financial regulations, which are a precondition for decision makers with respect to location & investment | country-specific regulations, condition of financial market, social considerations, environmental considerations | [45,68] | country rank on the ease-of-doing-business index: country rank < 75 (E3.1a) country rank 75–125 (E3.2a) country rank > 125 (E3.3a) |
| (17) financial support | financial support from political institutions for innovative projects such as loans, equity financing, or guarantees can incentivise RMs from mineral waste | active socio-political support | [102] | probability of approval: high (E3.1a) medium (E3.2a) low (E3.3a) |

**Table A4.** Environmental viability (E b) for the overall project rating with the UNFC-compliant categorisation matrix.

| Factor | Explanation | Dependence on | Modification after | Indicator & UNFC Rating |
|---|---|---|---|---|
| *Environmental impacts during project execution* | | | | |
| (18) air emission | risk of tailings being eroded by wind | particle size, TSF cover, local climate, wind conditions, pit configuration | [13,98] | risk of dust emission: low (<80%) (E1b) medium (50–80%) (E2b) high (>50%) (E3b) |
| (19) liquid effluent emission | effluents from tailings can contaminate soil & surface water | soil liner, drainage system, wet tailings storage, local environment, tailings' chemical properties | [13,98] | risk of groundwater contamination: low (E1b) medium (E2b) high (E3b) |
| (20) noise emission | noise & vibrations during mining; transport & processing can cause disturbances of local communities determined by individual & collective perception | mine planning, protective measures, topography, proximity to human settlements | [97] | expected degree of impact: low (E1b) medium (E2b) high (E3b) |
| *Environmental impacts after project execution* | | | | |
| (21) biodiversity | influence on habitats & species | local ecosystem, mining system, landscape, rehabilitation measures | [97] | total number of protected species that are affected by mining activities & that will be resettled on post-mining land: all (100%) (E1b) some (1–99%) (E2b) none (0%) (E3b) |
| (22) land use | land requirement after mine closure | amount of new residues, type of disposal, rehabilitation, land development opportunities | [97] | freely available post-mining land: most (>80%) (E1b) some (50–80%) (E2b) little (<50%) (E3b) |
| (23) material reactivity | capability of contained minerals to produce AMD | target minerals, concentration of sulphidic minerals | [13,103] | reduction of reactive material's mass: high (>80%) (E1b) medium (50–80%) (E2b) low (<50%) (E3b) |

**Table A5.** Social viability (E c) for the overall project rating with the UNFC-compliant categorisation matrix.

| Factor | Explanation | Dependence on | Modification after | Indicator & UNFC Rating |
|---|---|---|---|---|
| *Social impacts during project execution* | | | | |
| (24) local community | commitment beyond formal regulatory requirements, the recognition of diverse values, & the right to be informed about issues & conditions that influence lives | communication with stakeholders, proximity to human urban, protected, or culturally relevant areas, participation of local communities in decision-making | [68,97,104] | probability of approval through active commitment: high (>80%) (E3.1c) medium (50–80%) (E3.2c) low (>50%) (E3.3c) |
| (25) health & safety | protection of workers & local communities from injuries & diseases, & environmentalpollution | mining system, local health & safety standards, corporate values for the establishment of a safe work environment & lively safety culture | [97] | total number of complaints or prosecutions for non-compliance in planning phase: none (plans have been communicated publicly) (E3.1.c) more than 1 (plans have been communicated publicly) (E3.2c) none (plans have not been communicated publicly) (E3.3c) |
| (26) human rights & business ethics | degree to which a mining company values ethically correct behaviour | wages, right to organise trade unions, bribery & corruption, violation of human rights, forcefully gained control over land, a country's governance | [97] | total number of complaints or prosecutions for non-compliance in planning phase: none (plans have been communicated publicly) (E3.1.c) more than 1 (plans have been communicated publicly) (E3.2c) none (plans have not been communicated publicly) (E3.3c) |
| *Social impacts due to project execution* | | | | |
| (27) wealth distribution | distribution of earning between mining company, local communities, & government | a country's governance, choice of suppliers, & contractors; percentage of locally hired workers; wages | [97] | total number of complaints or prosecutions for non-compliance in planning phase: none (plans have been communicated publicly) (E3.1.c) more than 1 (plans have been communicated publicly) (E3.2c) none (plans have not been communicated publicly) (E3.3c) |
| (28) investment in local human capital | fostering personal skill development & capacity-building of employees by education & skill development | percentage of locally hired workers, offering higher education & training & transferable skill development; degree to which work is contracted out | [97] | percentage of employees sourced from local communities: high (>80%) (E3.1c) medium (50–80%) (E3.2c) low (<50%) or unclarified (E3.3c) |
| (29) degree of RM recovery | RMs can become inaccessible for recovery for future generations | disposal of new residues, mineral processing, residue stabilisation, residue characteristics | - | residue disposal: complete residue valorisation (E1c) separate disposal (E3.1c) mixed disposal (E3.2c) sterilisation (E3.3c) |
| (30) RM valorisation | utilising a RM in a sustainable manner to limit the impact of its recovery on the environment | target minerals, maturity of valorisation technologies, potential markets, RMs prices | [97] | total mass reduction as percentage of original tailings mass: high (>80%) (E1c) medium (50–80% (E2c) low (<50%) (E3c) |
| *Social impacts after project execution* | | | | |
| (31) aftercare | level of commitment & necessary measures on post-mining land | land management, national regulations, rehabilitation measures | - | duration of aftercare measures: short-term (<5 years) (E1c) mid-term (5–30 years) (E2c) long-term (>30 years) (E3c) |
| (32) landscape | mining activities can cause a visual impact by transforming landscapes | topography, local ecosystem, mine planning, local climate | [97] | impact on the environment: positive (E1c) neutral (E2c) negative (E3c) |

**Table A6.** Legal viability (E d) for the overall project rating with the UNFC-compliant categorisation matrix.

| Factor | Explanation | Dependence on | Modification after | Indicator & UNFC Rating |
|---|---|---|---|---|
| *Legal situation (relevant for project development)* | | | | |
| (33) right of mining | regulations affecting project planning & realisation | supranational, national, & regional laws & rules | [45] | state of development: application in development (E3.1d) / authorities engaged (E3.2d) / application not begun or unclarified (E3.3d) |
| (34) environmental protection | regulations affecting project planning & realisation | supranational, national, & regional laws & rules | [45,53,97] | state of development: application in development (E3.1d) / authorities engaged (E3.2d) / application not begun or unclarified (E3.3d) |
| (35) water protection | regulations affecting project planning & realisation | supranational, national & regional laws & rules | [45] | state of development: application in development (E3.1d) / authorities engaged (E3.2d) / application not begun or unclarified (E3.3d) |

**Table A7.** Degree of confidence in the geological estimates (G) for the rating of individual RMs with the UNFC-compliant categorisation matrix.

| Factor | Explanation | Dependence on | Modification after | Indicator & UNFC Rating |
|---|---|---|---|---|
| *Geological situation (relevant for project development)* | | | | |
| (36) quantity | amount of target RMs | ore quality, former processing efficiency, deposit volume | [45] | degree of geological certainty: high (G1) / medium (G2) / low (G3) |
| (37) quality | physico-chemical properties of target RMs | former processing, potential revenues | [45] | degree of geological certainty: high (G1) / medium (G2) / low (G3) |
| (38) homogeneity | distribution of target RMs inside the deposit | mine planning, mineral feed grade, timing of revenues | [45] | degree of geological certainty: high (G1) / medium (G2) / low (G3) |

**Table A8.** Technical feasibility (F) for the rating of individual RMs with the UNFC-compliant categorisation matrix.

| Factor | Explanation | Dependence on | Modification after | Indicator & UNFC Rating |
|---|---|---|---|---|
| *Mine planning considerations (relevant for project execution)* | | | | |
| (39) recoverability | ability to extract a wanted RM from the tailings | technological development, state of metallurgical testing, equipment availability, state of target RM | - | percentage of RM which is extracted from the tailings: high (>80%) (F1) / medium (50–80%) (F2) / low (<50%) (F3) |

**Table A9.** Economic viability (E a) for the rating of individual RMs with the UNFC-compliant categorisation matrix.

| Factor | Explanation | Dependence on | Modification after | Indicator & UNFC Rating |
|---|---|---|---|---|
| *Microeconomic aspects (relevant for project development)* | | | | |
| (40) demand | existence of a current practical use for the RM & absence of geological, technological, economic, environmental, social, &/or legal objections against its recovery | market, price, available technology, public acceptance, regulations | - | favourable conditions for RM extraction:<br>yes (E3.1a)<br>conditionally (E3.2a)<br>no (E3.3a) |
| (41) RM criticality | importance of a RM in an industry or economy | economic importance, supply risk, substitutability | [59] | allocation to EC's criticality assessment:<br>CRM (E1a)<br>high economic importance or supply risk (E2a)<br>no criticality (E3a) |
| (42) price development | forecasted RM price behaviour | demand, supply risk, quality, & quantity of historical data | - | forecasted mean price development over the project's duration:<br>positive trend (E3.1a)<br>stagnant trend (E3.2a)<br>negative trend (E3.3a) |

**Table A10.** Environmental viability (E b) for the rating of individual RMs with the UNFC-compliant categorisation matrix.

| Factor | Explanation | Dependence on | Modification after | Indicator & UNFC Rating |
|---|---|---|---|---|
| *Impacts after project execution* | | | | |
| (43) solid matter | a RM's potential to harm human health, flora, &/or fauna | concentration, toxicity, valorisation path | [13,105,106] | concentration of RM solid matter in new residues to qualify for class DK 0 (inert waste) according to German Landfill Regulation DepV [61]:<br>non-hazardous material (E1a)<br>threshold value not exceeded (E3.1a)<br>threshold value exceeded (E3.2a)<br>unclarified (E3.3a) |
| (44) eluate | a RM's potential to harm human health, flora, &/or fauna | concentration, toxicity, valorisation path, solubility | [13,105,106] | concentration of RM in eluate from new residues to qualify for class DK 0 (inert waste) according to German Landfill Regulation DepV [61]:<br>non-hazardous material (E1a)<br>threshold value not exceeded (E3.1a)<br>threshold value exceeded (E3.2a)<br>unclarified (E3.3a) |

**Table A11.** Knowledge base on the Bollrich tailings deposit for project definition. The dark grey shaded fields indicate data associated with high uncertainties, while the light grey shaded fields indicate data associated with moderate uncertainties, and the dashes indicate factors for which no information is available.

| Category & Factor | Data | Sources | UNFC Axis [1] |
|---|---|---|---|
| **(A) type of study** | very preliminary study | - | |
| **(B) basic information** | | | |
| (a) geography | | | |
| (i) location | Goslar district, Lower Saxony (Germany) (51°54′8.97″ N, 10°27′47.31″ E), 270 m above mean sea level nearest human settlement ~400 m E air-line distance downstream of main dam | [50] | |
| (ii) topography | at the foot of Harz mountain range, up to 1141 m altitude with deep valleys | [107] | |
| (iii) local geology | folded & faulted Paleozoic rocks of the Harz Mountains are uplifted & thrust over younger Mesozoic rocks of the Harz foreland along the Northern Harz Boundary fault leading to steeply tilting & partly inverted Mesozoic strata; Mesozoic rocks are largely composed of Triassic to Cretaceous sedimentary rocks of varying composition (i.e., mostly impure limestones, clastic sandstones (greywackes) & shales); younger Quaternary sediments are rare & locally limited | [108] | |

**Table A11.** *Cont.*

| Category & Factor | Data | Sources | UNFC Axis [1] |
|---|---|---|---|
| (iv) land use | in near vicinity: agricultural, forest, industrial & commercial, & recreation & residential areas | observed on Google Earth [50] | |
| (v) surface waters | Four small rivers observed downstream of TSF within a 1.5 km radius (Abzucht, Ammentalbach, Gelmke & Oker) | observed on Google Earth [50] | |
| (vi) climate | moderately warm, temperature $-0.7$ to $16.3\,^{\circ}\text{C}$ (average $7.2\,^{\circ}\text{C}$), average rain precipitation 911 mm/a, average climatic water balance 366 mm/a | [109,110] | |
| (b) geogenic deposit | | | |
| (i) mineralisation | two strongly deformed lens-shaped main ore bodies (high & low grade), sedimentary exhalative deposit (SedEx), fine grained (10–30 μm) principle sulphide minerals sphalerite ($(\text{Zn},\text{Fe})\text{S}$) & pyrite ($\text{FeS}_2$), less amounts of galena ($\text{PbS}$) & chalcopyrite ($\text{CuFeS}_2$), Ag, Au, (average estimated grades 14 wt% Zn, 6 wt% Pb, 2 wt% Cu, 140 g/t Ag & 1 g/t Au), barite ($\text{BaSO}_4$) (average grade 20 wt%)—additionally ca. 30 trace elements such as Co, Ga, & In, hosted by Middle Devonian Wissenbach shales | [50,107,111] | |
| (ii) former mining | underground mine, closed for economic reasons in 1988 after >1000 years of operation, now UNESCO World Heritage site located ~3 km W air-line distance from second processing plant Bollrich & TSF | [50,107,111] | |
| (c) tailings deposit | | | |
| (i) data collection methods | scientific publications or publicly accessible data, assumptions based on scientific publications, &/or own reasoning | - | |
| (ii) history | was in operation for ~49 years, decommissioned in 1987; supplied by processing plants Rammelsberg (into upper pond, 1938–1987) & Bollrich (into lower pond, 1956–1987); course of river Gelmke was changed several times | [53,57,107] | |
| (iii) recoverability | | | |
| • target minerals | previously & non-previously mined minerals | - | G |
| • quantity & quality | $V_\text{tailings} = 2{,}030{,}000\ \text{m}^3$, $m_\text{dry} = 7{,}100{,}000\ \text{t}$, $\rho = 3.5\ \text{t/m}^3$ (weighted mean value), $\rho_\text{neutralisation sludge} = 2.3\ \text{t/m}^3$ | [53,54] | |
| | exploration of deposit: (i) 10 drill cores (17–28 m) taken in upper pond along main dam & parallel to main dam in the middle of the pond, analysis of 16 elements; (ii) 90 water depth metering points | [53] | G |
| | 26 drill cores taken in upper & lower ponds, analysis of 4 elements & 3 minerals | [54] | |
| | low degree of alteration associated with oxidation | [53] | |
| • TSF structure | valley impoundment, estimated surface area 315,000 m$^3$<br>consists of 3 ponds: (i) lower pond (west, 74 vol% of TSF, $\rho = 3.0\ \text{t/m}^3$, max. water depth 4 m, average water depth 2 m), (ii) upper pond (middle, 26 vol% of TSF, $\rho = 3.7\ \text{t/m}^3$, max. water depth 0.5 m, average water depth 0.4 m), (iii) water retention pond (East)<br>consists of 3 dams: (i) main dam (max. 33 m height, max. 18° slope, raised 6 times, up-stream), (ii) middle dam (max. 19 m height), (iii) water retention dam (max. 8 m height) | [53,66], Ruler Tool [50], average water depth estimated with data from Reference [53] | F |
| • homogeneity | drill core data of upper pond shows relatively homogeneous deposit with slightly increasing Ba grades with depth; deposit modelled based on historical & current terrain models, water depth measurements, historical & current core data; validation by comparison to production records | [53] | G, F |
| • safety considerations | dam stability: occurrence of sinkhole at northern part of TSF documented in 1986 & several sinkholes near TSF reported in the past, which are associated with karstified geological structures nearby; expertise from 1986 concludes that TSF is not imminently threatened; confirmed by current calculations; unexploded ordnance: existence of WWII [2] ordnance cannot be excluded based on historical data so it needs to be investigated prior to mining | [53] | F |
| (iv) rehabilitation | not rehabilitated, left to ecological succession, no signs of AMD [3] or erosion observable | [53], observed on Google Earth [50] | |

**Table A11.** *Cont.*

| Category & Factor | Data | Sources | UNFC Axis [1] |
|---|---|---|---|
| (v) assessment status | | | |
| • maturity level | research work | - | |
| • characterisation | complete for lower pond | [53] | |
| | partial for upper pond; not all elements/minerals analysed; amount, composition, & shape of deposition of mine water neutralisation sludge in upper & lower pond roughly estimated | | |
| • evaluation | partial | - | |
| • classification | prospective project (E3F3G4) | [43] | |
| (vi) economics | | | |
| • RM criticality | $BaSO_4$, Co, Ga, & In are CRMs in EU with very high economic importance; Cu, Pb, & Zn have high economic importance in EU | [112] | E a |
| • further valorisation | industrial & metalliferous minerals of interest, use of residues in construction materials conceivable | - | E a |
| (vii) social impacts | | | |
| • health protection | no apparent imminent hazards known; negative impacts through dermal contact, ingestion or inhalation not given; risk assessment not performed | [53] | E c |
| • scientific interest | first scientific exploration shortly in 1983 before TSF abandonment in 1988; one recent research project (REWITA) with focus on mineral RMs recovery (2015–2018); proposal for follow-up project (REMINTA) on material extraction submitted | [53,54], www.cutec.de/fileadmin/Cutec/documents/cutec-news/2020/new58_dezember2020.pdf (accessed on 24 February 2021) | E c |
| • SLO [4] | positive perception of project idea by administrative bodies, environmental NGOs, & scientists | [52] | E c |
| | local population's perception of project idea unknown | - | |
| (viii) environmental impacts | | | |
| • pollution | possible negative impacts unknown; disused landfill "Paradiesgrund" located 250 m N air-line distance from TSF; possible influence on landfill when mining the TSF needs to be investigated | [53] | E b |
| | TSF's base not sealed & in direct contact with tailings | | |
| • landscape | integrated into landscape (visible only from up close or from hills); environment has been adapting through natural succession; active gilder airfield ~100 m N air-line distance from TSF; hiking trails next to TSF & biking Euroroute R1 near TSF | cf., Figure 2 | E b |
| • current status | on-site inspection of the TSF showed that rare flora, & aerial & soil fauna colonise the site | [53] | E b |
| • protected areas | conservation areas & protected landscapes nearby, protected species of flora & fauna sighted in area around TSF | [53] | E b |
| • secondary use | since 1966, neutralised mine water from the Rammelsberg mine has been discharged into the TSF (mainly upper pond, currently ~450,000 to 900,000 $m^3$/a); overlay of tailings and neutralisation sludge | [54] | E b |

**Table A11.** *Cont.*

| Category & Factor | Data | Sources | UNFC Axis [1] |
|---|---|---|---|
| **(d) technology** | | | |
| (i) mine planning | mine planning considerations on conceptual basis (dredging) | - | F |
| (ii) processing | extraction of $BaSO_4$, Co, Cu, Ga, In, Pb, Zn, & inert residues evaluated in discontinuous laboratory experiments on tailings from lower pond, processing sequences: (i) sulphide separation together with contaminants (rougher+cleaner+leaching), (ii) $BaSO_4$ separation (rougher+cleaner+scavenger+conditioning); recovery rates (tested on material from lower pond; ammonia leaching route for sulphides): $BaSO_4$ (74%), Co (12%), Cu (74%), Ga (2%), In (26%), Pb (65%), Zn (72%) & inert material (93%) processing tests on tailings from upper pond not performed; precipitation of $SO_4$ ions in multiple stages necessary to recover metals | [60] | F |
| **(e) infrastructure** | | | |
| (i) real estate | buildings & land from former processing available | [53] | F |
| (ii) mining & processing | former processing plant available ~550 m E air-line distance from TSF | [53] | |
| (iii) utilities | access to public electricity, gas, & water grid assumed | based on observation on Google Earth [50] | F |
| (iv) transportation & access | dirt roads, federal highway B6 ~1.6 km N air-line distance from TSF & public railway ~500 m E air-line distance from TSF; disused railway tracks from processing plant Bollrich to public network (estimated abandonment in 1988) | [53], observed on Google Earth [50] | F |
| **(f) politics** | | | |
| (i) political willingness | - | - | E c |
| **(g) legislation/licensing** | | | |
| (i) ownership | Bergbau Goslar GmbH (address: Bergtal 18, 38640 Goslar, Germany) | [53] | E d |
| (ii) legal exploration framework | currently supervised under German Federal Mining Act (BBergG) | [53] | E d |
| (iii) legal mining framework | - | - | E d |
| (iv) operating license | - | - | E d |
| (v) contracts | - | - | E d |
| **(C) mineral- & material-centric information** | | | |
| **(a) chemical & mineralogical composition** | | | |
| (i) elements | Ba (14.4), Cu (0.15), Fe (12.5), Pb (1.2), Zn (1.3) [mean, wt%]; Ag (-), As (700), Cd (30), Co (185), Ga (23), In (5.9), Tl (70) [mean, μg/g] | [53] | G |
| (ii) minerals | | | G |
| • main mineral groups (& associated elements) | silica-based: Al, Si, K, Ni, Ga carbonate: Ca, Mn, Fe, (Mg), (Co) sulphidic: Fe, Co, Cu, Zn, Pb, As, Cd, In, Tl sulphate: Ba, Ca | [53,54] | |
| • quantities: | estimated cumulated minerals content (total dry mass/share of tailings' mass) | [53] | |
| • $BaSO_4$ | 1,739,000 t/24.5 wt% (monomineralic) | | |
| • $CuFeS_2$ | 31,000 t/0.44 wt% | | |
| • $FeS_2$ | 1,086,000 t/15.3 wt% (7.1 wt% Fe in tailings) | | |
| • PbS | 85,000 t/1.2 wt% | | |
| • ZnS | 149,000 t/2.1 wt% | | |

**Table A11.** *Cont.*

| Category & Factor | Data | Sources | UNFC Axis [1] |
|---|---|---|---|
| • Wissenbach shales | 2,350,000 t/33.1 wt% | | |
| • ankerit | 1,611,000 t/22.7 wt% | | |
| • main minerals in neutralisation sludge: | masses unknown; high & low concentrations of Zn & $BaSO_4$, respectively | [54] | |
| • carbonate | $CaCO_3$ | | |
| • clay minerals | $Al_2O_3$ | | |
| • zinc hydroxide | $Zn(OH)_2$ | | |
| • quartz | $SiO_2$ | | |
| • gypsum | $CaSO_4 \cdot 2\,H_2O$ | | |
| (b) physico-chemical properties | | | |
| • particle size distribution | tailings: very fine, 90% of particles < 60 μm, predominantly 2–60 μm & partially >20% below 3 μm, analysed with 4 samples from 2 drill coresneutralisation sludge: very fine, ~80% of particles < 20 μm | [53,54] | G |
| • geomechanical properties | classified into geomechanical category GK III according to DIN 1054: highly difficult regarding the interaction of structure & subsoil | [113] | G |
| • abrasiveness | expected to be abrasive (30 wt% abrasive material in tailings) | [53] | G |
| • water content | 29 wt%, estimated mean water content | [53] | G |
| • toxic elements | no valorisation as soil possible due to heavy metal concentration (As, Cd, Cr, Cu, Hg, Ni, Pb, Tl, & Zn) according to guideline "LAGA TR Boden" (note: tailings are not soil per definition); classified as DK IV hazardous waste according to Landfill Regulation DepV; As, Cd, & Tl mainly associated with sulphides (As mainly with $FeS_2$ & Cd mainly with ZnS) | [53,114] | G |

[1] econ.: economic aspects, env.: environmental aspects, soc.: social aspects, leg.: legal aspects. [2] WWII: Word War II. [3] AMD: acid mine drainage. [4] SLO: social license to operate.

**Table A12.** Basic data for the in-situ rehabilitation scenario NRR0.

| Parameter | Unit | Value | Source | Remarks |
|---|---|---|---|---|
| surface area | $m^2$ | 315,000 | estimated with Google Earth [50] | - |
| duration of closure & leachate phase | a | 5 | following scenario B in Reference [51] (p. 104) | leachate emission constant; influx assumed only to occur in closure phase until in-situ stabilisation is completed & influx of rainwater or groundwater is phase neglected |
| duration of aftercare phase | a | 30 | Landfill Ordinance DepV [61] | minimum duration according to Landfill Ordinance DepV [61] |
| average emission of leachate | $m^3/a$ | 39,000 | average water depth for lower & upper ponds calculated based on 82 out of 90 measurements taken from Reference [53]; visible water surface measured with Google Earth [50] | based on the assumption of a constant leachate flow & that only the standing water is drained |
| leachate treatment | - | - | assumption | active on-site treatment unit |

**Table A13.** Economic parameters for closure and aftercare in the in-situ stabilisation and rehabilitation scenario NRR0. A conversion rate GBP-EUR of 0.9 is assumed as per 14 August 2020 [115] and rounded up. From the referenced sources, the maximum values are chosen for a conservative approach.

| Parameter | Unit | Value | Source | Remarks |
|---|---|---|---|---|
| **In-situ Stabilisation & Surface Sealing** | | | | |
| final surface cover including infrastructure | $€/m^2$ | 100 | [51] | closure & leachate phase |
| concrete injection | $€/m^3$ | 68 | [69] (p. 77) | closure & leachate phase |
| **Leachate treatment** | | | | |
| active on-site treatment | $€/m^3$ | 50 | [51] | closure & leachate phase |
| **Other Costs** | | | | |
| maintenance & repair of leachate collection system | $€/(a\ m^2)$ | 0.6 | [51] | closure & leachate phase |
| monitoring of leachates | $€/(a\ m^2)$ | 0.4 | [51] | closure & leachate phase |
| monitoring of groundwater | $€/(a\ m^2)$ | 0.3 | [51] | closure & leachate phase |
| insurances | $€/(a\ m^2)$ | 0.4 | [51] | closure & leachate phase |
| maintenance of surface sealing | $€/(a\ m^2)$ | 1.0 | [51] | aftercare phase |
| maintenance of infrastructure | $€/(a\ m^2)$ | 0.6 | [51] | aftercare phase |
| monitoring of settlement | $€/(a\ m^2)$ | 0.1 | [51] | aftercare phase |
| monitoring of environment including weather | $€/(a\ m^2)$ | 0.2 | [51] | aftercare phase |
| aftercare management, reports, & documentation | $€/(a\ m^2)$ | 0.6 | [51] | aftercare phase |

**Table A14.** Fixed economic and technological parameters for the techno-economic assessment of the mineral RMs recovery scenarios CRR1 and ERR2. A conversion rate USD–EUR of 0.85 is assumed as per 4 August 2020 [116].

| Parameter | Unit | Value | Source | Remarks | Qty. |
|---|---|---|---|---|---|
| **CAPEX** | | | | | |
| *Mining* | | | | | |
| dredger (including cutterhead) | € | 1,579,000 | [117] (p. SU 12), www.cat.com/en_US/ products/new/power-systems/ marine-power-systems/commercial-propulsion-engines/18493267.html (accessed on 14 March 2021) | 230 kW ship engine (d) [1], 272 kW cutterhead (d–e) [2], Caterpillar C18 ACERT engine used as reference | 1 |
| excavator | € | 160,000 | www.cat.com/en_US/products/new/ equipment/excavators/medium-excavators/1000032601.html (accessed on 14 March 2021) | CAT 320 GC, 1 $m^3$ bucket capacity, (d) | 1 |
| wheel loader | € | 269,000 | [117] (p. SU 22) | 157 kW (d), 3.8 $m^3$ bucket capacity | 1 |
| bulldozer (with ripper) | € | 145,000 | [117] (p. SU 28) | - | 1 |
| dump truck | € | 384,000 | [117] (p. SU 34) | 6x6 traction, 15 $m^3$ loading capacity, (d) | 1 |
| rubber boat (incl. engine) | € | 4800 | www.marine-sales.de (accessed on 14 March 2021) | transport of crew & light material to dredger, (d) | 2 |
| twin silo (2 × 810 $m^3$) | € | 343,000 | [117] (p. Misc 92) | ensuring continuous processing plant feed & contingency for feed stream disruptions; integrated stirring function assumed to keep tailings suspended | 1 |
| slurry pump | € | 24,000 | [117] (p. misc 56) | 41 kW (e) [3], 40 m head @ 90 $m^3$/h, redundant system foreseen | 6 |

**Table A14.** *Cont.*

| Parameter | Unit | Value | Source | Remarks | Qty. |
|---|---|---|---|---|---|
| pipeline | €/m | 1350 | [118] (p. 42) | 300 mm nominal diameter, assumed to be suitable for offshore & onshore application; 800 m one-way, redundant system foreseen; water recirculation included | 267 |
| floating bodies for pipeline | € | 8750 | [118] (p. 46) | longest distance to cover from landing site at northern part of middle dam to bottom right corner of lower dam (480 m) | 40 |
| *Processing* | | | | | |
| processing plant reactivation | € | 6,000,000 | [119] (p. 13) | low value is chosen since assets & machinery were assumed to be in place & reusable | - |
| *Infrastructure* | | | | | |
| mine site development (paving roads, reactivating railway, etc.) | € | 1,300,000 | [119] (p. 13) | low value chosen due to simple mine plan, good mine site accessibility & available buildings | - |
| *reclamation* | | | | - | |
| removal of assets, surface rehabilitation, & environmental monitoring | €/$t_{tailings}$ | 2 | [101] (p. 117) | mean value assumed due to relatively small reclamation area & off-site residue disposal | - |
| **Other Fixed Economic Parameters** | | | | | |
| discount rate | % | 15 | [8] (p. 297) | low value chosen to reflect very high risk | - |
| contingency factor | % | 30 | [45] (p. 58) | accounts for required non-specified assets | - |
| liquidating value | % | 10 | [120] (p. 16) | applied to assets & machinery under mining to estimate residual value | - |
| mine life | a | 11 | estimated with Taylor's Rule [62] (p. 80) | reclamation & asset liquidation only in year 11 | - |
| run-of-mine (ROM) | t/h | 170 | assumption | - | - |
| working days administration | d/a | 260 | assumption | - | - |
| working days mining | d/a | 260 | assumption | - | - |
| working days processing | d/a | 365 | assumption | - | - |
| shift system mining | shifts/d | 2 | assumption | 8 h per shift | - |
| shift system processing | shifts/d | 3 | assumption | 8 h per shift | - |
| working hours administration | h/d | 8 | assumption | - | - |
| working hours mining | h/d | 16 | assumption | - | - |
| working hours processing | h/d | 24 | assumption | - | - |
| %-NSR$_{Cu}$ (Europe) | % | 65 | [62] (p. 75) | percentage of net smelter return for Cu | - |
| %-NSR$_{Pb}$ | % | 65 | [62] (p. 75) | percentage of net smelter return for Pb | - |
| %-NSR$_{Zn}$ | % | 50 | [62] (p. 75) | percentage of net smelter return for Zn | - |

**Table A14.** *Cont.*

| Parameter | Unit | Value | Source | Remarks | Qty. |
|---|---|---|---|---|---|
| **Technological Parameters** | | | | | |
| tailings mass | t | 7,100,000 | [53] (p. AP1/75) | low value chosen for conservative approach | - |
| pump head | m | 55 | [53] (p. AP5/19) | - | - |
| $r_{Ba}$ [4] | % | 74 | [60] (p. 254) | - | - |
| $r_{Co}$ | % | 12 | [60] (p. 254) | for ammonia leaching path of sulphides | - |
| $r_{Cu}$ | % | 74 | [60] (p. 176) | - | - |
| $r_{FeS2}$ | % | 87 | [60] (p. 176) | - | - |
| $r_{Ga}$ | % | 2 | [60] (p. 254) | for ammonia leaching path of sulphides | - |
| $r_{In}$ | % | 26 | [60] (p. 254) | for ammonia leaching path of sulphides | - |
| $r_{inert\ material}$ | % | 93 | [60] (p. 254) | - | - |
| $r_{Pb}$ | % | 68 | [60] (p. 176) | - | - |
| $r_{Zn}$ | % | 70 | [60] (p. 176) | - | - |

[1] (d): diesel engine. [2] (d–e) diesel-electric engine. [3] (e): electric engine. [4] r: recovery rate.

**Table A15.** Variable economic parameters for the techno-economic assessment of the mineral RMs recovery scenarios CRR1 and ERR2. A conversion rate USD–EUR of 0.85 are assumed as per 4 August 2020 [116]. Data adopted from Reference [117] if not stated otherwise.

| Machine/Item | Energy Consumption [$l_{diesel}$/h] | Energy Consumption [$kW_{electricity}$] | Maintenance & Overhaul [€/h] | Remarks |
|---|---|---|---|---|
| dredger | 125 | - | 112 | fuel consumption @ 502 kW approximated based on specification sheet & CAT engine assumed to constantly deliver 502 kW, http://s7d2.scene7.com/is/content/Caterpillar/LEHM0004-00 (accessed on 15 March 2021) |
| excavator | 13 | - | 13 | - |
| wheel loader | 24 | - | 20 | - |
| bulldozer (with ripper) | 21 | - | 16 | - |
| dump truck | 15 | - | 13 | - |
| rubber boat (including engine) | 2 | - | - | no data could be retrieved for maintenance & overhaul, negligible due to expected low value |
| twin silo ($2 \times 810\ m^3$) | - | - | 5.8 | - |
| slurry pump | - | 41 | 3.2 | - |

**Table A16.** Variable economic parameters for the techno-economic assessment of the mineral RMs recovery scenarios CRR1 and ERR2. A conversion rate USD–EUR of 0.85 is assumed as per 4 August 2020 [116] if not stated otherwise.

| Parameter | Unit | Value | Source | Remarks | Qty. |
|---|---|---|---|---|---|
| **OPEX** | | | | | |
| *mining* | | | | | |
| machine operating costs | €/h | 200 | derived from Reference [117] | overhaul, maintenance, lubricants, & wear | - |
| diesel consumption | l/h | 202 | derived from Reference [117] | - | - |
| electric energy consumption | kW | 246 | derived from Reference [117] | - | - |
| shift supervisor | €/(a person) | 78.4 | based on Reference [120] | including assumed employers' share of 40% | 2 |
| machine driver | €/(a person) | 58.8 | based on Reference [120] | including assumed employers' share of 40% | 10 |
| metal worker | €/(a person) | 70.0 | based on Reference [120] | including assumed employers' share of 40% | 2 |
| *processing* | | | | | |
| processing costs | €/$t_{metal\ recovered}$ | 7.2 | [119] | - | - |
| machine operating costs | €/$t_{metal\ recovered}$ | 10.7 | [119] | electric energy only | - |
| shift supervisor | €/(a person) | 78.4 | [120] | including assumed employers' share of 40% | 3 |
| control panel operator | €/(a person) | 58.8 | [120] | including assumed employers' share of 40% | 3 |
| machine operator | €/(a person) | 58.8 | [120] | including assumed employers' share of 40% | 3 |
| metal worker | €/(a person) | 70.0 | [120] | including assumed employers' share of 40% | 3 |
| *services & administration* | | | | | |
| general services | €/d | 5210 | [119] | - | - |
| administrative services | €/d | 1310 | [119] | - | - |
| **RM prices** | | | | | |
| electricity | €/kWh | cf., Figure S1 | raw data from Reference [85] | forecast based on yearly average prices in Germany for commercial customers from 2014–2019 | - |
| diesel | €/l | cf., Figure S2 | raw data from Reference [86] | forecast based on yearly average prices in Germany from 1950–2020 | - |
| BaSO$_4$ | €/$t_{tailings}$ | cf., Figure S3 | raw data from References [87–90] | forecast based on yearly BaSO$_4$ prices from 2011–2020 [1] | - |
| Co | €/$t_{tailings}$ | cf., Figure S4 | raw data from References [87,89–93] | forecast based on yearly Co prices from 1996–2020 [1] | - |
| Cu | €/$t_{tailings}$ | cf., Figure S5 | raw data from Reference [94] | forecast based on monthly Cu prices from November 1999–March 2021 [1] & price per tonne tailings estimated after Wellmer et al. [62] (p. 47 ff.) | - |
| Ga | €/$t_{tailings}$ | cf., Figure S6 | raw data from References [87,89–93] | forecast based on yearly Ga prices from 1999–2020 [1] | - |
| In | €/$t_{tailings}$ | cf., Figure S7 | raw data from References [87,89–93] | forecast based on yearly In prices from 1999–2020 [1] | - |
| Pb | €/$t_{tailings}$ | cf., Figure S8 | raw data from Reference [95] | forecast based on monthly Pb prices from November 1999–March 2021 [1] & price per tonne tailings estimated after Wellmer et al. [62] (p. 74 ff.) | - |
| Zn | €/$t_{tailings}$ | cf., Figure S9 | raw data from Reference [96] | forecast based on monthly Zn prices from November 1999–March 2021 [1] & price per tonne tailings estimated after Wellmer et al. [62] (p. 74 ff.) | - |
| residue sales | €/t | 5.0 | assumption | intended valorisation as filler in construction materials; reference value for high-quality sand in Goslar is EUR 19.5 (www.recyclingpark.de/startseite.html, accessed on 2 June 2021); lower price assumed to estimate conservatively due to lack of information on effort to condition residues | - |
| residue disposal | €/t | 40.0 | [53] (p. AP7-9/58) | high value chosen to estimate conservatively | - |

[1] under consideration of monthly/yearly USD–EUR conversion rates.

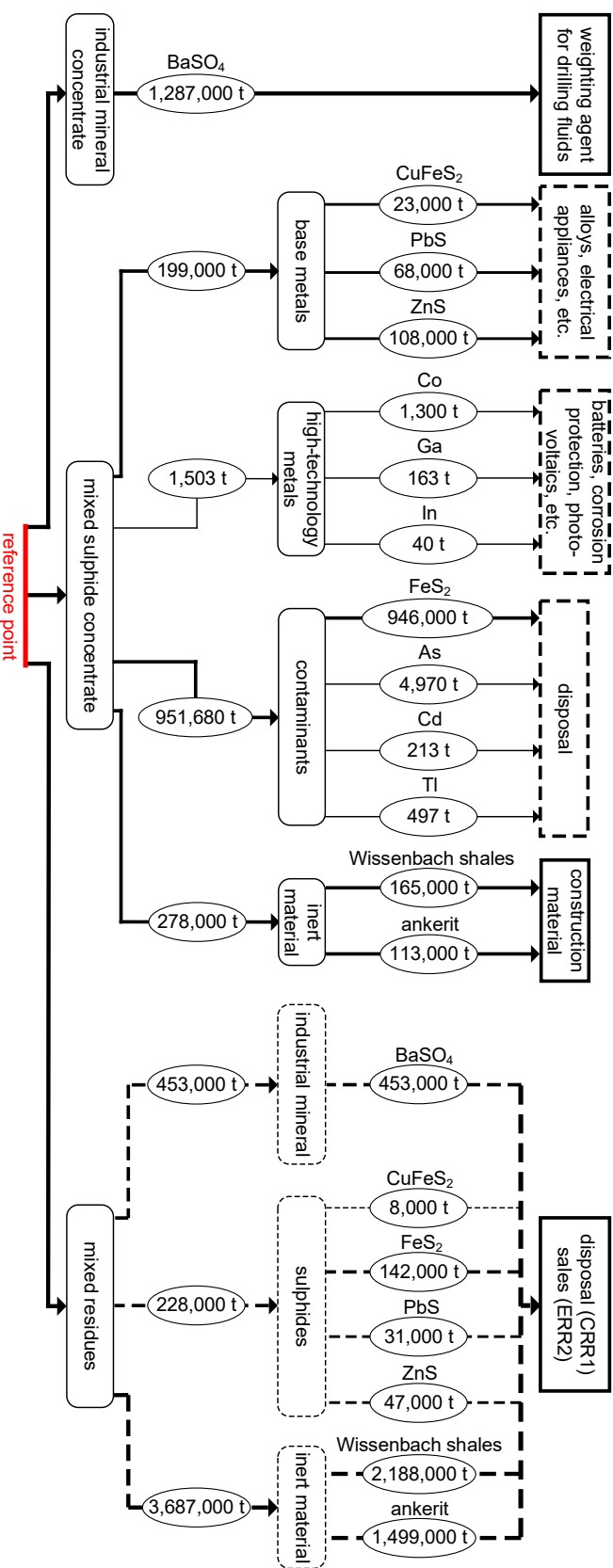

**Figure A1.** Detailed production breakdown of 10-year material flows for the RMs recovery scenarios (CRR1, ERR2).

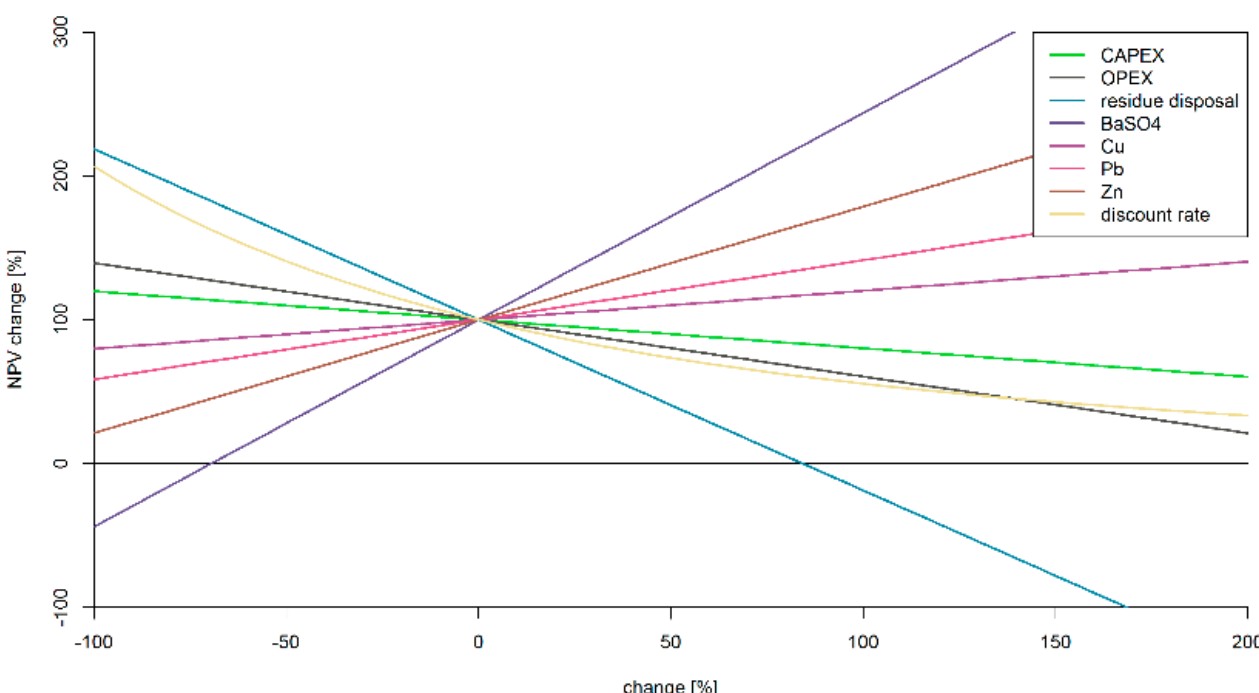

**Figure A2.** Results of the sensitivity analysis of the conventional mineral RMs recovery scenario (CRR1m) with mean price forecast and a discount rate of 15%.

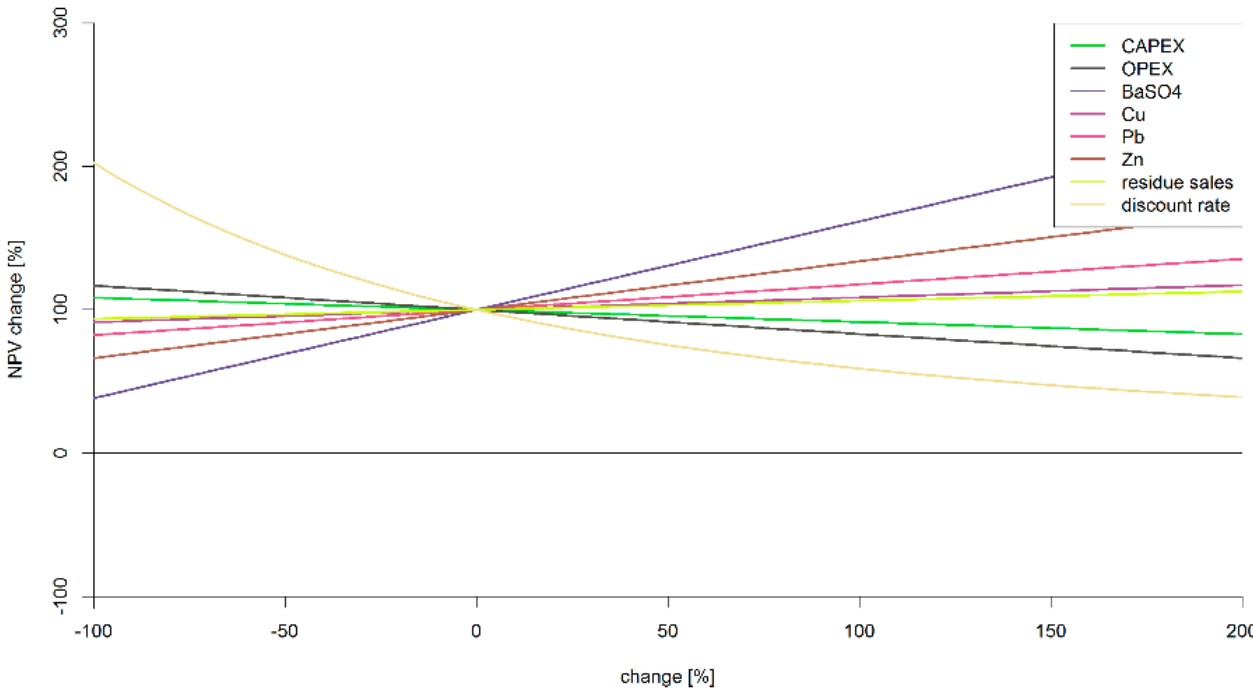

**Figure A3.** Results of the sensitivity analysis of the enhanced mineral RMs recovery scenario (ERR2m) with mean price forecast and a discount rate of 15%.

**Table A17.** Overall project rating with the UNFC-compliant categorisation matrix of the degree of confidence in the geological estimates (G).

| Factor | Indicator | UNFC Rating | Justification | Source |
|---|---|---|---|---|
| *Geological conditions (relevant for project development)* | | | | |
| (1) quantity | degree of geological certainty: | | | |
| | medium | G2 | NRR0, CRR1, & ERR2: deposit modelled based on direct data on 10 drill cores from lower pond, and pre-processed historical data on 14 & 12 drill cores from lower & upper pond, respectively. Model was validated with historical production data. Extension & volume of TSF known with medium confidence. Overall knowledge on mineral quantity with medium confidence in both ponds. Knowledge gap on quantity of neutralisation sludge & other dumped material. | [53] |
| (2) quality | degree of geological certainty: | | | |
| | medium | G2 | NRR0, CRR1, & ERR2: physico-chemical properties known with medium confidence. | [53] |
| (3) homogeneity | degree of geological certainty: | | | |
| | medium | G2 | NRR0, CRR1, & ERR2: mineral distribution in lower pond known with medium confidence. Knowledge gap on distribution of tailings & neutralisation sludge in both ponds. | [53,54] |

**Table A18.** Overall project rating with the UNFC-compliant categorisation matrix for the technical feasibility (F).

| Factor | Indicator | UNFC Rating | Justification | Source |
|---|---|---|---|---|
| *TSF condition & risks (relevant for project development)* | | | | |
| (4) ordnance | degree of knowledge: | | | |
| | unclarified | F3 | NRR0, CRR1, & ERR2: existence cannot be excluded based on historical data. Requires clarification. | [53] |
| *Mine planning considerations (relevant for project execution)* | | | | |
| (5) mine/ operational design | level of detail of planning: | | | |
| | basic | F3 | NRR0, CRR1, & ERR2: conceptual planning. | - |
| (6) metallurgical testwork | degree of research on mineral processing: | | | |
| | - | - | NRR0: factor not applicable. | - |
| | laboratory scale | F3 | CRR1 & ERR2: extraction of $BaSO_4$, Co, Cu, Ga, In, Pb, Zn, & inert material (Wissenbach shales, ankerit) evaluated in discontinuous laboratory experiments on tailings from lower pond. | [60] |
| (7) water consumption | percentage of recycled water: | | | |
| | high (>80%) | F1 | CRR1 & ERR2: water recirculated in dredging operation. Processing water can be recirculated, too. | [53] |
| | unclarified | F3 | NRR0: unclear if TSF water can be used for making concrete. | - |

**Table A18.** *Cont.*

| Factor | Indicator | UNFC Rating | Justification | Source |
|---|---|---|---|---|
| *Infrastructure (relevant for project development)* | | | | |
| (8) real estate | condition of infrastructure: | | | |
| | highly developed | F1 | NRR0, CRR1, & ERR2: buildings & land from former processing available. | [53] |
| (9) mining & processing | condition of equipment: | | | |
| | - | - | NRR0: not applicable since specialised non-mining equipment is required. | - |
| | bleak | F3 | CRR1 & ERR2: unclarified. | - |
| (10) utilities | condition of infrastructure: | | | |
| | acceptable | F2 | NRR0, CRR1, & ERR2: access to public electricity, gas, & water grid assumed. | based on observation on Google Earth [50] |
| (11) transportation & access | condition of infrastructure: | | | |
| | acceptable | F2 | NRR0, CRR1, & ERR2: dirt roads, federal highway B6 ~1.6 km N air-line distance from TSF & public railway ~500 m E air-line distance from TSF, disused railway tracks from processing plant Bollrich to public network (estimated abandonment in 1988). | [53], observed on Google Earth [50] |
| *Post-mining state (relevant for future impacts)* | | | | |
| (12) residue storage safety | suitability of new disposal site for safe storage: | | | |
| | unclarified | F3 | NRR0: predicting long-term stability might be difficult. CRR1 & ERR2: new disposal site unknown. | [69] |
| (13) rehabilitation | level of detail of planning: | | | |
| | conceptual | F2 | NRR0, CRR1, & ERR2: conceptual planning. | - |

**Table A19.** Overall project rating with the UNFC-compliant categorisation matrix of the economic viability (E a).

| Factor | Indicator | UNFC Rating | Justification | Source |
|---|---|---|---|---|
| *Microeconomic aspects (relevant for project development)* | | | | |
| (14) economic viability | discounted cash flow over projected LOM: | | | |
| | positive (NPV >> 0€) | E3.1a | CRR1m & ERR2m: NPVs of EUR 73 mio. & EUR 172 mio., respectively, with mean price forecast. | - |
| | negative (NPV << 0€) | E3.3a | NRR0: costs of EUR 125 mio. incurred. | - |
| (15) economic uncertainty | uncertainty of cash flow in pessimistic scenario: | | | |
| | - | - | NRR0: no forecast performed. | - |
| | low (NPV in pessimistic scenario >> 0€) | E3.1a | ERR2p: NPV = EUR 73 mio. | - |
| | high (NPV in pessimistic scenario << 0€) | E3.3a | CRR1p: NPV = EUR −17 mio. | - |

**Table A19.** *Cont.*

| Factor | Indicator | UNFC Rating | Justification | Source |
|---|---|---|---|---|
| *Financial aspects (relevant for project development)* | | | | |
| (16) investment conditions | country rank in the ease-of-doing-business Index. | | | |
| | - | - | NRR0: not applicable since company works on assignment basis. | - |
| | high (<75) | E3.1a | CRR1 & ERR2: country rank 22 (Germany). Good investment conditions assumed. | [121] |
| (17) financial support | probability of approval: | | | |
| | high | E3.1a | CRR1 & ERR2: research on TSF was funded publicly & positive results give rise to the assumption that follow-up project proposal REWIMET might be accepted. | - |
| | no financial support scheme available | E3.3a | NRR0: no financial support scheme known at the moment. | - |

**Table A20.** Overall project rating with the UNFC-compliant categorisation matrix for the environmental viability (E b).

| Factor | Indicator | UNFC Rating | Justification | Source |
|---|---|---|---|---|
| *Environmental impacts during project execution* | | | | |
| (18) air emission | risk of dust emission: | | | |
| | unclarified | E3.3b | NRR0: unclarified if TSF needs to be drained prior to concrete injection, which could lead to wind erosion of the tailings. | - |
| | high (>80%) | E3.1b | CRR1 & ERR2: complete submersion of tailings in dredging operation. | - |
| (19) liquid effluent emission | risk of groundwater contamination: | | | |
| | low | E3.1b | NRR0, CRR1, & ERR2: status quo is expected to be retained. | - |
| (20) noise emission | expected degree of impact: | | | |
| | medium | E3.2b | NRR0, CRR1, & ERR2: constant noise emission from TSF in 2 working shifts from Mondays to Fridays. Noise is expected to be audible, especially in the surrounding mountain area & areas on the same plane. It is possible that the noise would not be audible in residential areas to topography. | based on observation on Google Earth [50] |
| *Environmental impacts after projection execution* | | | CRR1 & ERR2: the processing plant is to be soundproofed. | |
| (21) biodiversity | total number of protected species that are affected by mining activities & that will be resettled on post-mining land: | | | |
| | none (0%) | E3b | NRR0, CRR1, & ERR2: protected flora & fauna species were sighted during an on-site inspection. Capturing the exact types & number of species is required for planning a resettlement or other compensation measures. | [53] |
| (22) land use | freely available post-mining land: | | | |
| | some (50–80%) | E3.2b | NRR0: surface area of current wet cover is made available for reuse. CRR1 & ERR2: original topography is restored. NRR0, CRR1, & ERR2: it is expected that a solution for the collection & further treatment of the neutralisation sludge requires a permanent land use. | - |
| (23) material reactivity | reduction in reactive material's mass: | | | |
| | high (>80%) | E3.1b | CRR1: 84 wt% of sulphides leave the system boundaries as commodities. ERR2: all tailings are valorised. | - |
| | low (<50%) | E3.3b | NRR0: factually, reactive materials remain in place. Long-term stability difficult to predict. | [69] |

**Table A21.** Overall project rating with the UNFC-compliant categorisation matrix for the social viability (E c).

| Factor | Indicator | UNFC Rating | Justification | Source |
|---|---|---|---|---|
| *Social impacts during project execution* | | | | |
| (24) local community | probability of approval through active commitment: | | | |
| | medium (50–80%) | E3.2c | CRR1 & ERR2: first indication of positive prospects by stakeholder assessment (local government, industry, university, & environmental NGOs). Local population's opinion unknown. | [52] |
| | unclarified | E3.3c | NRR0: no data available. | - |
| (25) health & safety | total number of complaints or prosecutions for non-compliance in planning phase: | | | |
| | none | E3.3c | NRR0, CRR1, & ERR2: plans have not been communicated publicly. | - |
| (26) human rights & business ethics | total number of complaints or prosecutions for non-compliance in planning phase: | | | |
| | none | E3.3c | NRR0, CRR1, & ERR2: plans have not been communicated publicly. | - |
| *Social impacts due to project execution* | | | | |
| (27) wealth distribution | total number of complaints or prosecutions for non-compliance in planning phase: | | | |
| | none | E3.3c | NRR0, CRR1, & ERR2: plans have not been communicated publicly. | - |
| (28) investment in local human capital | percentage of employees sourced from local communities: | | | |
| | unclarified | E3.3c | NRR0: it can be expected that an external contractor must be hired due to the special character of the required services. Aftercare measures could be carried out by local workers. CRR1 & ERR2: unclarified how many local workers could be employed. | - |
| (29) degree of RM recovery | residue disposal: | | | |
| | complete residue valorisation | E1c | ERR2: no loss since all tailings are valorised. | - |
| | mixed disposal | E3.2c | CRR1: it is assumed that the site for the disposal of new residues has no option to store different residues separately. | - |
| | sterilisation | E3.3c | NRR0: access to RM potential for future generations with reasonable effort prevented. | - |
| (30) RM valorisation | total mass reduction as percentage of original tailings mass: | | | |
| | high (>80%) | E3.1c | ERR2: all tailings are valorised. | - |
| | low (<50%) | E3.3c | NRR0: no valorisation takes place. CRR1: 38 wt% of tailings are valorised. | - |
| *Social impacts after project execution* | | | | |
| (31) aftercare | duration of aftercare measures: | | | |
| | short-term (up to 5 years) | E1c | CRR1 & ERR2: aftercare assumed to be complete after 1 year | - |
| | long-term (more than 30 years) | E3c | NRR0: long-term behaviour difficult to predict & long-term monitoring might be necessary. | [69] |
| (32) landscape | impact on the environment: non-perceptible | E1c | CRR1 & ERR2: former topography is restored. | - |
| | partially perceptible | E2c | NRR0: is expected to be well integrated into landscape with an according surface design. Main dam remains perceptible. | - |

**Table A22.** Overall project rating with the UNFC-compliant categorisation matrix for the legal viability (E d).

| Factor | Indicator | UNFC Rating | Justification | Source |
|---|---|---|---|---|
| *Legal situation (relevant for project development)* | | | | |
| (33) right of mining | state of development: application not begun or unclarified | E3.3d | NRR0, CRR1, & ERR2: no concrete activities initiated. | - |
| (34) environmental protection | state of development: application not begun or unclarified | E3.3d | NRR0, CRR1, & ERR2: no concrete activities initiated. | - |
| (35) water protection | state of development: application not begun or unclarified | E3.3d | NRR0, CRR1, & ERR2: no concrete activities initiated. | - |

**Table A23.** Rating of individual RMs with the UNFC-compliant categorisation matrix for the degree of confidence in the geological estimates (G).

| Factor | Indicator | UNFC Rating | Justification | Source |
|---|---|---|---|---|
| *Geological conditions (relevant for project development)* | | | | |
| (36) quantity | degree of geological certainty: | | | |
| | medium | G2 | CRR1 & ERR2: knowledge on $BaSO_4$, Cu, $FeS_2$, Pb, Zn, & inert material (Wissenbach shales, ankerit) with medium confidence in both ponds. | [53,54] |
| | low | G3 | CRR1 & ERR2: knowledge on Co, Ga, & In with medium confidence in lower pond. Co, Ga, & In quantity in upper pond inferred. | [53] |
| (37) quality | degree of geological certainty: | | | |
| | medium | G2 | CRR1 & ERR2: knowledge on $BaSO_4$, Cu, $FeS_2$, Pb, Zn, & inert material (Wissenbach shales, ankerit) with medium confidence in both ponds. | [53,54] |
| | low | G3 | CRR1 & ERR2: knowledge on Co, Ga, & In with medium confidence in lower pond. Co, Ga, & In quantity in upper pond inferred. | [53] |
| (38) homogeneity | degree of geological certainty: | | | |
| | medium | G2 | CRR1 & ERR2: knowledge on the distribution of $BaSO_4$, Cu, $FeS_2$, Pb, Zn, & inert material (Wissenbach shales, ankerit) with medium confidence. | [53,54] |
| | low | G3 | CRR1 & ERR2: knowledge on the distribution of Co, Ga, & In with medium confidence in lower pond. Knowledge on Co, Ga, & In in upper pond inferred. | [53] |

**Table A24.** Rating of individual RMs with the UNFC-compliant categorisation matrix for the technical feasibility (F).

| Factor | Indicator | UNFC Rating | Justification | Source |
|---|---|---|---|---|
| *Mine planning considerations (relevant for project execution)* | | | | |
| (39) recoverability | percentage of RM which is extracted from the tailings: | | | |
| | high (>80%) | F1 | CRR1 & ERR2: $FeS_2$ (87 wt% recovered in mixed sulphide concentrate), inert material (Wissenbach shales, ankerit) (93 wt% are recovered with the new residues). | [60] |
| | medium (50–80%) | F2 | CRR1 & ERR2: $BaSO_4$ (74 wt%), Cu (74 wt%), Pb (68 wt%), Zn (70 wt%). | [60] |
| | low (>50%) | F3 | CRR1, ERR2: Co (12 wt%), Ga (2 wt%), In (26 wt%). | [60] |

**Table A25.** Rating of individual RMs with the UNFC-compliant categorisation matrix for the economic viability (E a).

| Factor | Indicator | UNFC Rating | Justification | Source |
|---|---|---|---|---|
| *Microeconomic aspects (relevant for project development)* | | | | |
| | favourable conditions for RM extraction: | | | |
| | yes | E3.1a | CRR1 & ERR2: there is a demand for $BaSO_4$, Cu, Pb, Zn, Co, Ga, & In | [122] |
| (40) demand | conditionally | E3.2a | CRR1 & ERR2: Fe & $H_2SO_4$ could theoretically be produced from $CuFeS_2$ & $FeS_2$. | [123] |
| | no | E3.3a | CRR1 & ERR2: residues theoretically usable in construction materials, but experiments are necessary. Currently, there is per se not a demand for residues so that a potential application of the inert fraction (Wissenbach shales, ankerit) of the new residues needs to be clarified. | - |
| | allocation to EC's criticality assessment: | | | |
| | CRM | E1a | CRR1 & ERR2: $BaSO_4$, Co, Ga, & In. | [112] |
| (41) RM criticality | high economic importance or supply risk | E2a | CRR1 & ERR2: Cu, Pb, S (from $CuFeS_2$ & $FeS_2$), & Zn. | [112] |
| | no criticality | E3a | CRR1 & ERR2: inert material (Wissenbach shales, ankerit). | |
| | forecasted mean price development over the project's duration: | | | |
| | - | - | CRR1 & ERR2: $FeS_2$ is recovered as a non-paid co-product, & no price forecast was performed for the inert material (Wissenbach shales, ankerit). | - |
| (42) price development | positive trend | E3.1a | CRR1 & ERR2: $BaSO_4$, Co, In. | Figures S3, S4 and S7 |
| | stagnant trend | E3.2a | CRR1 & ERR2: Pb, Zn. | Figures S8 and S9 |
| | negative trend | E3.3a | CRR1 & ERR2: Cu, Ga. | Figures S5 and S6 |

**Table A26.** Rating of individual RMs with the UNFC-compliant categorisation matrix for the environmental viability (E b).

| Factor | Indicator | UNFC Rating | Justification | Source |
|---|---|---|---|---|
| *Impacts after project execution* | | | | |
| | concentration of RM solid matter in new residues to qualify for class DK 0 (inert waste) according to German Landfill Regulation DepV [61]: | | | |
| (43) solid matter | - | - | NRR0: not applicable since no new residues are produced. ERR2: not applicable since no new residues are disposed of. | - |
| | non-hazardous material | E1b | CRR1 & ERR2: inert material (Wissenbach shales, ankerit). | - |
| | threshold value not exceeded | E3.1b | CRR1: Cu, Zn. | [60] |
| | threshold value exceeded | E3.2b | CRR1: Pb. | [60] |
| | concentration of RM in eluate from new residues to qualify for class DK 0 (inert waste) according to German Landfill Regulation DepV [61]: | | | |
| (44) eluate | - | - | NRR0: not applicable since no new residues are produced. ERR2: not applicable since no new residues are disposed of. | - |
| | non-hazardous material | E1b | CRR1 & ERR2: inert material (Wissenbach shales, ankerit). | - |
| | threshold value not exceeded | E3.1b | CRR1: Ba, Cu, Zn. | [60] |
| | threshold value exceeded | E3.2b | CRR1: Pb. | [60] |

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
