# Peer review of "How to Identify Potentials and Barriers of Raw Materials Recovery from Tailings? Part II: A Practical UNFC-Compliant Approach to Assess Project Sustainability with On-Site Exploration Data"

_resources, doi:10.3390/resources10110110_

Round 1

Reviewer 1 Report

The article proposes a structured standard methodology for a first appraisal concerning the feasibility of re-mining anthropogenic RMs existing in TSFs, in compliance with the United Nations Framework Classification for Resources. The methodology allows to identify potentials and barriers and intends to verify the possibility to reconcile different stakeholder interests (economic, environmental and social). The conceptual model is applied to a case study – the former Rammelsberg mining operation. A sensitivity and an uncertainty analysis are absolutely required. The proposed methodology enables a point by point comparison of different scenarios so that the most auspicious option can be quickly identified.

General appraisal:

  • Presentation and Clarity - Text relatively easy to read. Logical sequence of clear, intentional exposure directed to the main objective. References are extensive and appropriate;
  • Integration and coherence - establishes a logical and rational relationship between the different contents;
  • State of the Art Review - objective survey;
  • Research Methods – Appropriated to the elected case study.
  • Data Analysis – Tables and schemes allow an easy understanding of the descriptive text.
  • Discussion of results – Exempt and open appreciation.
  • The contribution to knowledge through the proposal of a potentially useful standardized methodology for re-mining anthropogenic RMs is unquestionable.

SUGGESTION

TABLE 2  - First line – it is written “for in-situ rehabilitation, TSF abandonment is performed as for DK II class landfills under the German Landfill Regulation (DepV) [61]”. A short footnote explaining what is a DK II class landfill would be very useful for the reader as most of them are not acquainted with German Legislation. Also the reference [61] points out to the Regulation text that is written in German.

Reviewer 2 Report

How to Identify Potentials and Barriers of Raw Materials  Recovery from Tailings? Part II: A Practical UNFC-Compliant  Approach to Assess Project Sustainability with On-Site  Exploration Data

 The authors also in the second part of the work maintain a high level of work, extremely necessary, issues very relevant to our planet , which is slowly becoming a global garbage can.  comprehensive multi-faceted approach,

This  Bollrich example can be a prelude to developing a resource management / recovery method for each landfill and tailings. The work presents a multi-threaded approach to identifying the potentials and barriers of recovery / recycling raw materials from waste generated by human activity.

The work can be an example of how to assess waste dumps in terms of management in accordance with the circular economy as well as risks associated with TSFs comprise the contamination of soil and water. The model and results are particularly suitable for use in post-production waste landfills containing basic to critical metals. Rich literature introduces the basic standardized procedure to explore tailings

I have only one comment:

Keywords: anthropogenic raw materials; Bollrich; critical raw materials; responsible raw material 29 sourcing; sustainability; tailings; United Nations Framework Classification for Resources (UNFC)

 Keywords are badly chosen, too broad, do not reflect the best conveyed content, three meaningful two-word keys are enough, after all, it is mainly about the sustainability assessment related to the recovery of recyclable materials,

\Why mention the name of the city /landfill Bollrich, the method/model should be universal because it concerns an increasingly cluttered world.

Why UNFC- advertising? in my opinion, it is enough if it is in the text of the work

 Please think about more universal keywords: secondary materials, recycling?

Article fulfills the conditions for printing in terms of technical and content.
